# KompeteAI: Accelerated Autonomous Multi-Agent System for End-to-End Pipeline Generation for Machine Learning Problems

## Abstract

Recent Large Language Model (LLM)-based AutoML systems demonstrate impressive capabilities but face significant limitations such as constrained exploration strategies and a severe execution bottleneck. Exploration is hindered by one-shot methods lacking diversity and Monte Carlo Tree Search (MCTS) approaches that fail to recombine strong partial solutions. The execution bottleneck arises from lengthy code validation cycles that stifle iterative refinement. To overcome these challenges, we introduce KompeteAI, a novel AutoML framework with dynamic solution space exploration. Unlike previous MCTS methods that treat ideas in isolation, KompeteAI introduces a merging stage that composes top candidates. We further expand the hypothesis space by integrating Retrieval-Augmented Generation (RAG), sourcing ideas from Kaggle notebooks and arXiv papers to incorporate real-world strategies. KompeteAI also addresses the execution bottleneck via a predictive scoring model and an accelerated debugging method, assessing solution potential using early-stage metrics to avoid costly full-code execution. This approach accelerates pipeline evaluation 6.9 times. KompeteAI outperforms leading methods (e.g., RD-agent, AIDE, and ML-Master) by an average of 3% on the primary AutoML benchmark, MLE-Bench. Additionally, we propose Kompete-bench to address limitations in MLE-Bench, where KompeteAI also achieves state-of-the-art results.

## 1 Introduction

Recent research has shifted towards the use of Large Language Models (LLM) as the core reasoning engine for AutoML frameworks, enabling the autonomous generation and testing of end-to-end pipelines that adapt to specific tasks (Li et al. (2024); Jiang et al. (2025); Chi et al. (2024); Liu et al. (2025); Trirat et al. (2024); Grosnit et al. (2024); Yang et al. (2025); Nam et al. (2025)). However, these approaches have critical limitations. Initial "one-shot" generation can yield diverse ideas but lacks iterative refinement, so a single flawed component - like suboptimal feature engineering — can undermine the entire pipeline without a way to correct it.

More adaptive frameworks based on Monte Carlo Tree Search (MCTS) address this by exploring a tree of potential solutions but often use constrained exploration and struggle to recombine promising ideas from different branches. Even recent frameworks that combine exploratory search with LLM-based reasoning Liu et al. (2025), despite achieving state-of-the-art results on MLE-Bench Chan et al. (2024), are limited by architectural constraints. They may not systematically preserve or merge valuable insights from distant high-performing branches during the exploration process. As a result, valuable solution components can be prematurely discarded.

Even perfect internal recombination only reshuffles known ideas. Retrieval-Augmented Generation (RAG) overcomes this by injecting external, domain-specific knowledge, enabling exploration beyond the pretrained hypothesis space. Yet current systems rarely exploit this potential: most apply RAG only in the early stages Trirat et al. (2024); Yang et al. (2025), causing retrieval to fail at providing fresh knowledge and leading to knowledge decay across the pipeline.

All frameworks also face a severe execution bottleneck. Validating a single solution requires full code execution, often taking hours. Debugging worsens this: late errors force complete retraining, slowing feedback and discouraging major changes. This creates a serious scalability challenge.

In this work, we introduce KompeteAI, an autonomous multi-agent framework for structured, multistage pipeline generation. The key innovations compared to prior approaches are presented in Table 1. To efficiently explore and exploit the search space, we employ two core operators. **Adding**, which dynamically generates novel stage-specific ideas by querying external knowledge sources via an **adaptive RAG module**, and **merging**, which intelligently combines the most successful solutions. We address execution bottlenecks through a **predictive scoring model** that prunes weak solutions early, alongside an accelerated debugging paradigm using simplified code and smaller data samples, dramatically shortening the feedback loop. These innovations accelerate the average test-time performance by a factor of 6.9.

Our experiments show that KompeteAI sets the new state-of-the-art on MLE-Bench, outperforming prior methods by an average of 3%.

Finally, the only publicly available competitive benchmark MLE-Bench suffers from two key limitations. It is excessively large, constructs its test sets by partitioning the original training data, and then compares the resulting scores to the private leaderboard positions that cause evaluation bias. To address these issues, we introduce a new benchmark — Kompete-bench.

Our primary contributions:

- **Stage-Decomposed Multi-Agent Architecture:** A state-of-the-art framework that partitions the ML workflow into discrete stages, enabling agents to specialize in focused tasks, dynamically integrate external knowledge sources to enhance exploration diversity, and systematically recombine optimal partial solutions through novel adding and merging operators.

- **An Accelerated Evaluation and Debugging Paradigm:** A two-part solution to the execution bottleneck, combining a predictive scoring model and a rapid debugging framework to drastically reduce validation time.

- **A New Benchmark:** Kompete-bench — A curated benchmark of recent real-world problems designed to more rigorously evaluate a model's genuine problem-solving ability, minimizing the influence of memorization or prior exposure.

## 2 RELATED WORK

**Classic AutoML.** Classic AutoML frameworks - TPOT, AutoGluon, AutoKeras, LightAutoML, and others Olson & Moore (2016); Erickson et al. (2020); Jin et al. (2019); Vakhrushev et al. (2021); Feurer et al. (2022); LeDell & Poirier (2020); Thornton et al. (2013) — automate data preprocessing, model selection, and hyperparameter tuning via heuristic search, ensembling, and Bayesian optimization. Despite their effectiveness, these systems operate within static search spaces, require manual data preparation and adaptation for each new task, and lack the dynamic coordination and continual learning capabilities inherent to multi-agent AutoML architectures.

**LLM-based AutoML.** LLM-based AutoML systems have progressed rapidly, offering increasingly autonomous capabilities through dynamic coordination and iterative planning (AutoKaggle Li et al. (2024), SELA Chi et al. (2024), AIDE Jiang et al. (2025), AutoML-Agent Trirat et al. (2024), RD-agentYang et al. (2025), ML-Master Liu et al. (2025), MLE-STAR Nam et al. (2025)). To analyze and compare these systems meaningfully, we focus on four key aspects. The results are presented in Table 1. Notably, RD-Agent's merging relies on uncontrolled LLM-driven recombination, often producing incoherent or suboptimal integrations, while MLE-STAR's single, terminal merging step simply consolidates ideas without meaningfully improving solution exploration (see Appendix C.2 for detailed analysis).

**Scoring Model.** Our approach of performance scoring is inspired by similar performance prediction methods developed in Neural Architecture Search Elsken et al. (2019). In NAS, a common approach is to use weight-sharing supernets, where multiple architectures are jointly trained by sharing parameters with a large model, and performance is estimated by evaluating sampled architectures on

| | Exploration | RAG | Debuging | Merging |
|---|---|---|---|---|
| **SELA** | MCTS (UCT) w/o context | ✗ | ✗ | ✗ |
| **AIDE** | MCTS (UCT) w/o context | ✗ | Incremental | ✗ |
| **AutoML-Agent** | Retrieval-augmented planning | R&D-phase | Incremental | ✗ |
| **R&D-Agent** | Multi-trace exploration | R&D-phase | Incremental | Simple Recombination |
| **ML-Master** | MCTS (UCT) with context | ✗ | Incremental | ✗ |
| **MLE-STAR** | Incremental exploration | ✗ | Single-stage | Final model ensembling |
| **KompeteAI** | Multi-stage expansion | Dynamic | Multistage | Controlled merger |

Table 1: Comparison of LLM-based AutoML systems. R&D-phase retrieval provides only a static, upfront knowledge injection. Simple recombination mixes candidate ideas without assessing their individual utility, while final model ensembling averages only end-stage models.

a validation set Jawahar et al. (2023a). This method, however, faces challenges such as weight co-adaptation Bender et al. (2018), capacity bottlenecks Jawahar et al. (2023b), and gradient conflicts Gong & Wang (2022). More recently Jawahar et al. (2023a), demonstrated that LLMs can serve as effective performance predictors in NAS tasks, providing a promising alternative to traditional methods. In contrast to NAS, the AutoML setting typically involves a much broader and less constrained search space, which motivates the exploration of LLM-based performance prediction beyond neural architectures.

**Benchmarks.** Using Kaggle competitions to evaluate autonomous ML systems has become a modern approach in a number of recent benchmarks due to clear metrics, variety of tasks, and the ability to compare with human solutions. One of the first benchmarks for systematically evaluating autonomous ML agents in Kaggle competitions was MLAgentBench Huang et al. (2023), which focused on a small set of tasks with simple baselines, measuring agents' ability to improve on them. Later, DSBench Jing et al. (2024), expanded the scope but often relied on automated filtering, which excluded many complex or non-standard competitions. The most recent effort, MLE-Bench Chan et al. (2024) stands out for its scale and diversity, presenting a more challenging and realistic testbed for multi-agent AutoML systems. However, MLE-Bench also faces notable limitations, including its large size (3.3 TB) and the fact that it constructs its test sets by partitioning the original training data and then compares the resulting scores to the private leaderboard positions that cause evaluation bias.

## 3 KOMPETEAI

The pipeline of KompeteAI is demonstrated in Figure 1. It is designed to ensure robust, leak-free data handling, efficient exploration of modeling ideas, and rapid iteration, all while maintaining high code quality and logical consistency. This section outlines the key components and mechanisms that underpin the operation of our system. It consists of three main stages: (3.1) Pipeline Setup, which prepares the core components; (3.2) Tree Initialization, which generates an initial set of candidate pipelines; and (3.3) Tree-Guided Exploration, the main stage involving tree operations such as node adding and merging, supported by a Scoring Model to speed up pipeline evaluation.

### 3.1 PIPELINE SETUP

This phase sets up the core components required for the next stages. The dataset is ingested by *The Reader Agent*, which analyzes its structure, produces a detailed task specification, and initializes the data for the RAG based on this description. *The Metric Agent* constructs unit tests to support submission validation and defines the evaluation metric function. *The Validator Agent* partitions the data according to the task specification and applies appropriate pre-processing methods to ensure a valid and reliable evaluation protocol. *The Baseliner Agent* generates an initial solution, establishing a lower-bound reference for expected performance. Based on the baseline score, it also assesses the quality of the data split and, if necessary, can trigger *The Validator Agent* to re-partition the data using an alternative strategy.

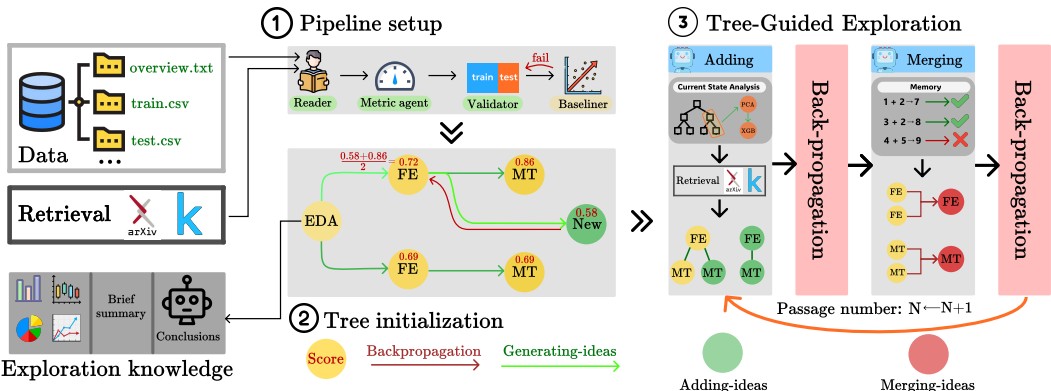

Figure 1: The KompeteAI AutoML pipeline.

## 3.2 TREE INITIALIZATION

The primary objective of the tree initialization stage is to generate an initial set of candidate pipelines, providing a diverse foundation from which further exploration can proceed. This stage seeds the search space with promising ideas and establishes the initial structure for subsequent optimization.

### 3.2.1 NODE REPRESENTATION

In our framework, each node encodes a code segment, and each tree level maps to a pipeline component: Exploratory Data Analysis (EDA), Feature Engineering (FE), or Model Training (MT). Connections between nodes define candidate pipelines. The root EDA node aggregates insights from data exploration — visualizations, distributions, and summary statistics — and can be dynamically enriched as the search progresses. In contrast, FE and MT nodes represent concrete instantiations of their phases, with edges marking their inclusion in the same solution. Each node may have multiple children but only one parent.

### 3.2.2 THE IDEATION PROCESS

This section focuses on the generation process for a single node within the pipeline, emphasizing localized decision-making rather than full-path construction. The procedure is structured as a modular interaction among four specialized agents. The Insighter Agent initiates the process by proposing candidate ideas. These are then evaluated by the Checker Agent for consistency and compatibility with the solution context. Once validated, the Coder Agent translates the idea into executable code. The output is again verified by the Checker Agent, this time to ensure implementation correctness. Finally, the Debugger Agent addresses potential runtime errors and integration issues, completing the node's initialization cycle.

**The Insighter Agent** generates ideas for downstream components, guided by factors such as EDA results and the current solution stage. To overcome LLM limitations in diversity and quality, it uses two key mechanisms.

- Tree Memory: The agent employs a memory module over the entire ideation tree, analyzing previously generated nodes using diversity-driven strategies based on embedding cosine similarity — such as selecting nodes closest or farthest from the parent, or sampling randomly. A top-$n$ subset is selected for inclusion in the context, and the memory is continuously updated as new nodes are added, enabling dynamic promotion of diversity.

- Retrieval-Augmented Generation: The RAG component retrieves high-quality ideas from Kaggle (top-$n$ similar tasks and winning solutions) and arXiv (top-$m$ papers returned by task-specific queries), forming a candidate pool of task-relevant concepts. Retrieved texts are then normalized and processed by the Insighter Agent to extract key ideas, which are

subsequently selected top-$k$ according to their contextual relevance and how well they complement the ideas already integrated into the working context. The retrieval approach is adaptive, based on function calling that is triggered only when external knowledge is expected to contribute meaningfully, thereby minimizing computational overhead. Notably, the RAG mechanism also supports the use of any pre-loaded text corpus, enabling the system to operate in restricted environments and to be tailored to highly specialized domains when required.

**The Checker Agent** is designed to validate the logical consistency of outputs produced by other agents. It uses unit tests, including schema validation and script execution, to assess correctness.

**The Coder Agent** implements the input idea based on several parameters, including a description of the input data, available computational resources, and the idea itself. It selects optimal hyperparameters and designs the architecture to ensure the code executes within the given time constraints while remaining as efficient as possible.

**The Debugger** agent efficiently resolves issues with dependency installation, code generation, and submission formatting using a nested loop that iteratively debugs code within a set limit. To reduce runtime overhead common in existing systems, it accelerates debugging by minimizing time-sensitive parameters like training iterations, enabling fast error detection without full execution. Once debugging succeeds, the original configuration is restored. The agent also logs detailed metrics — such as error types, retries, and outcomes — and skips debugging steps for recurring errors, opting instead for direct code regeneration.

### 3.3 TREE-GUIDED EXPLORATION

Tree-Guided Exploration has three components: Adding, Merging, and Scoring Model. The Tree Initialization stage builds the primary structure. Within the time budget, the system alternates between Adding — injecting new ideas into promising branches — and Merging, which combines two strong ideas into a single solution. The scoring model works alongside these phases to accelerate pipeline evaluation. After each Adding–Merging cycle, we backpropagate performance signals from MT nodes (where evaluation scores are available) upward, updating each node's average score to improve their representativeness in the search tree. The average score of each node is calculated as the mean of the scores of its children.

#### 3.3.1 ADDING

The adding stage governs the structured expansion of the ideation tree $T_t = (V_t, E_t)$ by proposing new nodes at the Feature Engineering and Model Training levels. Before expansion, the agent examines the current tree state and may selectively trigger additional exploratory analysis via function calling, choosing specific data inspections that are most informative for subsequent node generation.

The process is conditioned on a global context vector $\mathbf{c}_t$, representing the current pipeline state and external knowledge in structured form.

We define $\mathbf{c}_t = \phi_{\text{EDA}}(T_t) + \phi_{\text{reader}}(T_t) + \phi_{\text{ext}}(\texttt{QueryExternal}())$, where each mapping $\phi : \mathcal{X} \to \mathcal{C}$ converts noisy inputs into structured semantics in the shared context space $\mathcal{C}$. Here, $\phi_{\text{EDA}}$ encodes statistical/structural signals from exploratory analysis, $\phi_{\text{reader}}$ extracts insights from metadata and past solutions, and $\phi_{\text{ext}}$ adds knowledge from external sources such as arXiv or Kaggle.

Based on the aggregated context $\mathbf{c}_t$, the agent performs a structured expansion of the ideation tree in three successive steps. First, it samples a set of candidate FE nodes according to the context $\mathbf{c}_t$, where $\mathcal{V}_{\text{FE}}$ denotes the space of available FE transformations. Then, for each sampled FE node $v^{\text{FE}}$, the agent generates a corresponding set of MT nodes within the MT configuration space $\mathcal{V}_{\text{MT}}$. Finally, a subset of FE nodes from the current tree (including both existing and newly added ones) is selected based on their scores, and for each selected node additional MT nodes are appended as children.

The full procedure is detailed in Appendix 1.

### 3.3.2 MERGING

The merging stage enables the agent to consolidate multiple promising solutions at the Feature Engineering and Model Training levels, yielding stronger and more generalizable configurations.

The merging process unfolds as follows.

1. A set of FE node pairs $(v_i^{\text{FE}}, v_j^{\text{FE}})$ is sampled, excluding those present in the long-term memory buffer $\mathcal{M}_{\text{long}}$. Each valid pair is merged into a new node $v_{ij}^{\text{FE}} = \text{MergeFE}(v_i^{\text{FE}}, v_j^{\text{FE}})$ which recombines structural and statistical traits of its parents.

2. For each merged node $v_{ij}^{\text{FE}}$, a set of child MT nodes is generated from a conditional distribution. Then, for each parent FE node $v_i^{\text{FE}}$ and $v_j^{\text{FE}}$, the agent selects additional MT nodes from their respective subtrees. These are sampled stochastically with probabilities proportional to their scores: $u_k^{(i)} \sim \text{SampleTop}(v_i^{\text{FE}})$ The final child set of $v_{ij}^{\text{FE}}$ combines the freshly generated MT nodes and the resampled top performers from its parents.

3. Additionally, a subset $\mathcal{V}_{\text{FE}}^{\text{merge}} \subseteq V_t^{\text{FE}}$ is selected, and within each selected FE node, MT child pairs are merged using $u_{ij}^{\text{MT}} = \text{MergeMT}(u_i, u_j)$ to form stronger model configurations.

   To avoid redundant or destructive merges, the agent employs dual memory buffers. The short-term memory $\mathcal{M}_{\text{short}}$ temporarily stores failed merge attempts, while the long-term memory $\mathcal{M}_{\text{long}}$ permanently excludes repeatedly failing node pairs. Specifically, a pair that fails to produce a beneficial merge $\theta_{\text{fail}}$ times is promoted from $\mathcal{M}_{\text{short}}$ to $\mathcal{M}_{\text{long}}$, ensuring efficient resource allocation and adaptive learning.

Merging procedure detailed in Appendix 2.

### 3.3.3 SCORING MODEL

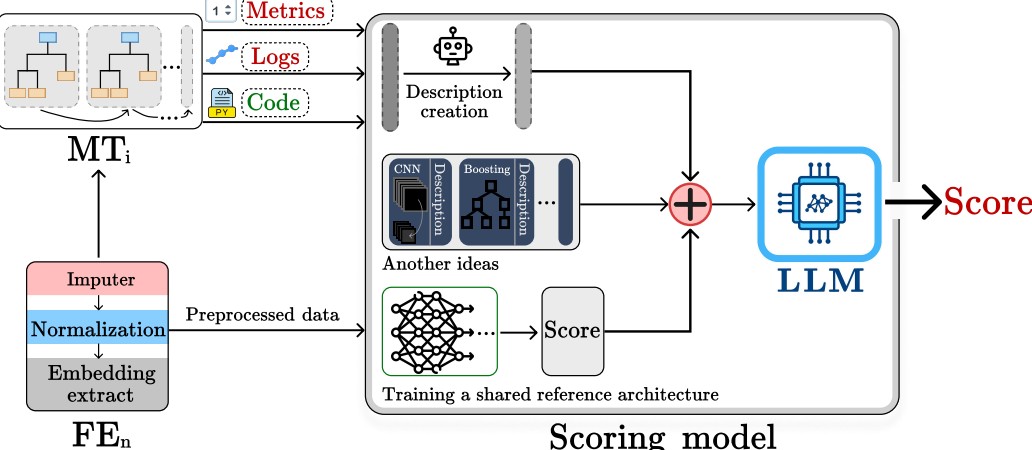

Figure 2: For each feature-engineering node FE$_n$, an accelerated model-training step MT$_i$ generates reduced-epoch logs, metrics, and a code-based description of the candidate. Simultaneously, a shared reference architecture is trained on the same FE$_n$ to capture preprocessing effects. These signals, combined with anchor ideas, are fed to the LLM, which predicts the final score without full training.

The scoring model is a crucial component applied at the model training nodes of our pipeline generation tree (Figure 2). Its primary purpose is to accelerate overall pipeline evaluation, enabling exploration of a greater number and diversity of candidate solutions. By providing rapid estimates of model performance, the scoring model allows prioritization of the most promising candidates without full-scale training, which can be prohibitively expensive for complex architectures. Formally, the predicted score $\hat{S}(m)$ of a candidate model $m$ is computed as:

$$\hat{S}(m) = f_{LLM}\Big( \underbrace{\phi_{\text{desc}}(m)}_{\text{MT}_i \text{ description}}, \underbrace{\phi_{\text{anchor}}(\{a_i\}_{i=1}^k)}_{\text{anchor ideas}}, \underbrace{\phi_{\text{metrics}}(S_{\text{ref}})}_{\text{reference architecture metrics}} \Big),$$

Each component of the scoring function captures a distinct aspect of the candidate evaluation process. The term $\phi_{\text{desc}}(m)$ encodes a detailed description of the candidate model, including its architecture, hyperparameters, and early-stage training behavior, such as reduced-epoch metrics and learning logs obtained from accelerated training. The anchor representation $\phi_{\text{anchor}}(\{a_i\})$ provides context from precomputed examples of similar models trained on the same dataset, allowing the scoring model to generalize performance patterns. Finally, $\phi_{\text{metrics}}(S_{\text{ref}})$ captures the effect of the data preprocessing at the current node by incorporating the performance of a shared reference architecture trained under the same feature-engineering configuration.

Anchor selection follows a systematic two-step procedure. First, multiple complex ideas — each representing a distinct model class — are trained on a baseline preprocessing setup to capture architectural variability. Second, a single shared reference architecture is trained on the current feature-engineering node to quantify the impact of that preprocessing choice on model performance. These signals, together with the candidate description and anchor ideas, are aggregated and passed to a large language model, which outputs a predicted final score without requiring full-scale training.

## 4 KOMPETE-BENCH

### 4.1 MOTIVATION

Despite growing interest in multi-agent AutoML, empirical evaluation remains limited by the lack of accessible and reliable benchmarks. Currently, the only public option is MLE-Bench — a collection of 75 Kaggle competitions that offer a rich mix of real-world ML challenges, clear evaluation metrics, and a competitive setup that closely mirrors practical deployments.

While MLE-Bench is a significant step forward, it has two key limitations that hinder its practical utility and reproducibility. First, it is prohibitively large: even the "Lite" version with just 22 competitions takes up 158 GB and requires substantial compute to process. Second, MLE-Bench constructs test sets by partitioning the original training data and compares results to private leaderboard positions. As shown in table 4, this introduces significant evaluation bias, as it does not reflect actual test set performance. We show that scores on MLE-Bench's constructed test sets can diverge significantly from real leaderboard rankings, leading to misleading conclusions and undermining its reliability as a proxy for real-world outcomes.

### 4.2 METHODOLOGY

We propose a concise two-part benchmark for evaluating multi-agent AutoML systems, balancing historical, contemporary, and future-oriented challenges under standardized computational constraints.

- **Selection of Established Competitions.** We curate 15 competitions from the 'lite' MLE-Bench collection that still accept late submissions on Kaggle and whose individual dataset sizes do not exceed 1 GB. The aggregate volume of this subset is 5.3 GB, providing a stable foundation for baseline comparisons.

- **Incorporation of New Competitions.** To better reflect the evolving landscape of AutoML tasks, we include 11 additional competitions from the years 2024 and 2025. These datasets, totaling 4.9 GB, were selected primarily to ensure fair comparison with human performance. This is because, given the recency of these competitions, both human participants and current models have access to almost the same tools and libraries, minimizing discrepancies due to technological advancements.

# 5 EXPERIMENT

## 5.1 EXPERIMENT SETUP

**Baselines.** To provide a comprehensive evaluation on MLE-Bench, we compared our system with the top open-source methods on the leaderboard: AIDE, RD-agent, ML-Master, and MLE-STAR. Due to the high cost of rerunning all baselines, we report their leaderboard metrics as provided by MLE-Bench. For Kompete-Bench, all methods were tested with the same LLM backend — gemini-2.5-flash Comanici et al. (2025), which also powers our system, except ML-Master, which was run on deepseek-r1 Guo et al. (2025) due to architectural constraints. To ensure a fair comparison between MLE-Bench and Kompete-Bench and eliminate potential bias from using different LLM backends, we further evaluated AIDE and RD-Agent with o1-preview Jaech et al. (2024) — the model on which most MLE-Bench submissions achieved their highest scores.

**Environment.** To evaluate our system on MLE-Bench, we set a 6-hour runtime limit using 12 vCPUs, 64 GB of RAM, and a single NVIDIA A100 GPU (40 GB). The same hardware setup was used for all systems on Kompete-bench, where we applied a stricter 6-hour time limit. Each configuration was executed three times, and results were averaged to reduce variance. To ensure fairness and prevent systems from exploiting test-time information, we restrict RAG components in each run from accessing any information about the competition solutions, including those released after its completion.

**Benchmarks.** We evaluate KompeteAI on the Lite subset of MLE-Bench, a computationally feasible proxy for the full benchmark. Although not identical, MLE-Bench reports high alignment in system rankings between the Lite and full versions, making Lite a practical basis for comparison. For medal-based evaluation against real Kaggle leaderboards, we include only Lite competitions that still accept submissions. Additionally, we assess performance on our custom benchmark — Kompete-bench, structured following the same principles described in our methodology section, by separating the data into two categories: MLE-subset Lite and recent and a collection of newly curated competitions.

**Evaluation Metrics.** On MLE-Bench, we adopt the official leaderboard metric — the percentage of submissions that receive a medal — which aligns with standard Kaggle competition criteria. For Kompete-bench, we additionally report the percent humans beaten metric, which measures the percentage of human participants outperformed by the agent on the corresponding Kaggle leaderboard. This metric offers a more fine-grained and interpretable evaluation signal compared to medal thresholds, allowing us to distinguish between systems that may not earn medals but differ meaningfully in their relative competitiveness against human participants. To assess pipeline efficiency, we report the number of complete pipeline passes, where a single pass corresponds to one adding and one merging iteration performed by the AutoML system.

## 5.2 RESULTS

**Evaluation on MLE-Bench.** We begin by evaluating our system on the widely adopted MLE-Bench benchmark (Lite subset). As presented in Table 2, KompeteAI attains the highest overall performance, outperforming existing state-of-the-art systems by an average margin of 3 percentage points.

| Name | MLE-Bench medals % |
|---|---|
| AIDE (o1-preview) | 34.3 ± 2.4 |
| RD-Agent (o1-preview) | 48.18 ± 1.1 |
| ML-Master (deepseek-r1) | 48.5 ± 1.5 |
| MLE-STAR (gemini-2.5-flash) | 43.9 ± 6.2 |
| KompeteAI (gemini-2.5-flash) | **51.5 ± 1.5** |

Table 2: Comparison of agents on MLE-Bench 'Lite' subset. For AIDE, RD-agent, MLE-STAR and ML-Master, we use results reported by MLE-Bench. KompeteAI was run 3 times with different seeds; results are reported as mean ± SEM. Each run was given 6 hours.

**Statistical significance** To more thoroughly assess the significance of our results on MLE-Bench, we first examined our system's confidence intervals, which do not overlap with those of competing baselines, indicating a statistically significant improvement. We further validated these findings using paired t-tests, with all p-values at or below 0.05, confirming that the observed gains are unlikely to arise by chance. Detailed results are shown in Table 3.

Table 3: Paired t-test results comparing our system to each baseline on MLE-Bench. All p-values ($p \leq 0.05$) indicate statistically significant performance gains over AIDE, RD-Agent, ML-Master, and MLE-Star.

| Name | AIDE | RD-Agent | ML-Master | MLE-Star |
|------|------|----------|-----------|----------|
| P-value | 0.00 | 0.02 | 0.04 | 0.05 |

**Kaggle-Based Validation of Robustness.** To validate robustness beyond proxy benchmarks, we tested all agents on the Lite subset using official Kaggle submissions (Table 4). The analysis reveals a substantial gap between MLE-Bench evaluations and actual leaderboard outcomes. In particular, all comparable systems exhibit inflated medal rates on MLE-Bench compared to their real Kaggle performance. This discrepancy highlights the limitations of relying solely on MLE-Bench for comparison, especially in competitive human-level settings.

| Name | MLE-Bench LB, % | Real LB, % | LBs medal intersection, % |
|------|-----------------|------------|---------------------------|
| AIDE (o1-preview) | 20 | 7 | 0 |
| RD-agent (o1-preview) | 27 | 13 | 7 |
| AIDE (gemini-2.5-flash) | 13 | 7 | 0 |
| RD-agent (gemini-2.5-flash) | 13 | 7 | 0 |
| ML-Master (deepseek-r1) | 27 | 13 | 7 |
| MLE-Star (gemini-2.5-flash) | 20 | 13 | 7 |

Table 4: Medal rate gap between the MLE-Bench leaderboard and the leaderboard of a real Kaggle competition. The "LBs medal intersection" column indicates the percentage of competitions where a medal was won on both the real LB and the MLE-Bench LB. For this experiment we ran each system once and used the rules from Kaggle to determine the medal.

**Evaluation on Kompete-Bench.** To address these limitations, Kompete-Bench evaluates performance using real Kaggle leaderboards, covering recent competitions with high prize pools and strong engagement. On this Contemporary subset, traditional medal-based evaluation fails, as all agents achieve zero medal rates. We therefore propose a finer-grained metric: *percent humans beaten*. As shown in Figure 3, KompeteAI demonstrates state-of-the-art results, surpassing human performance in 14% of cases — significantly ahead of all comparable systems. Nevertheless, the agent remains far behind top leaderboard teams, reflecting how contemporary competitions demand capabilities beyond local modeling — including large-scale feature engineering, synthetic data generation, and use of external datasets. Human participants leverage broader tools and resources, underscoring the competitiveness of these challenges and the remaining headroom for automated agents.

**Impact of the Acceleration Methods.** Finally, we assess the contribution of our acceleration framework. Table 5 reports the number of iterations within a fixed budget. By integrating predictive scoring and an accelerated debugging loop, our pipeline executes over 6.9× more iterations than a baseline without accelerations, enabling stronger submissions under identical constraints. Moreover, as shown in Figure 3, KompeteAI without accelerations exhibits a marked performance drop, underscoring its ability to refine solutions during test-time optimization. While we restrict acceleration measurements to KompeteAI due to tight pipeline integration, the proposed paradigms are general and may be adapted to other architectures with minimal conceptual changes.

## 5.3 ABLATION STUDY

As shown in Figure 3, all major components of KompeteAI are crucial, with their impact particularly pronounced on the Contemporary subset.

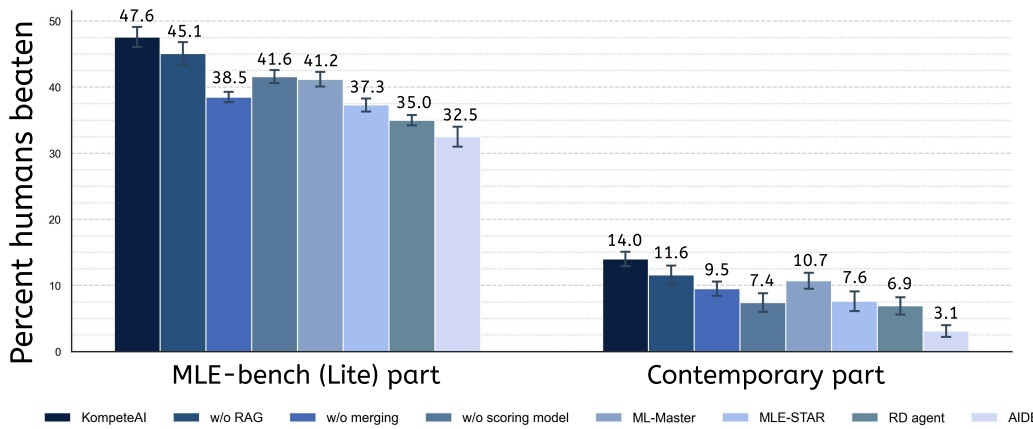

Figure 3: Comparison of our pipeline with AIDE, RD-agent and ML-Master on Contemporary and MLE-Bench parts of Kompete-bench. All systems use gemini-2.5-flash as the underlying LLM, except for ML-Master, which uses deepseek-r1. Each was run 3 times with different seeds, results are averaged, and each run was limited to 6 hours.

| System Configuration | Number of Iterations |
|---|---|
| w\o all accelerations | 1.8 ± 0.3 |
| w\o scoring model | 4.1 ± 0.4 |
| with all accelerations | **12.5 ± 4.1** |

Table 5: Impact of acceleration techniques on the number of completed iterations within a fixed time budget. Mean ± 95% CI estimated via Student's $t$-distribution.

- **W\o RAG** The removal of RAG causes a sharper *relative* decline on Contemporary tasks: performance drops from 14% to 11.6%, compared to a more moderate reduction from 47.6% to 45.1% on MLE-Bench Lite. This indicates that in recent competitions, producing strong solutions in isolation is far harder, and the ability to incorporate external ideas and tools becomes disproportionately important.

- **W\o Merging** The merging mechanism yields one of the largest absolute improvements. Without it, performance falls to 9.5% on Contemporary and 38.5% on MLE-Bench Lite. While the agent can still generate diverse ideas, strong final submissions typically emerge only when partial yet promising solutions are consolidated.

- **W\o Scoring model** Removal of the scoring model lowers results to 7.4% on Contemporary and 41.6% on MLE-Bench Lite. By prioritizing high-potential candidates, the scoring model enables broader exploration, allowing the agent to test significantly more hypotheses under limited compute.

# 6 CONCLUSIONS

We introduced KompeteAI, a new multi-agent AutoML framework, and demonstrated its strong performance on diverse and challenging tasks. Our contributions include a state-of-the-art architecture with dynamic integration of external knowledge sources and novel tree-based operators for adding and merging, combining LLM decision-making with algorithmic methods. We further propose an Accelerated Evaluation and Debugging Paradigm to address execution bottlenecks, applicable to general LLM-based AutoML architectures. Finally, we show that the current benchmark suffers from significant bias and introduce Kompete-Bench to overcome this limitation.

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

## A  FUTURE WORK

Our work opens several promising directions for future research.

First, while our acceleration paradigm significantly reduces computation time, it relies on a scoring model whose accuracy may degrade over longer runs, potentially leading to cumulative errors. Improving this component — e.g., through adaptive retraining or uncertainty-aware corrections — could enhance long-term robustness.

Second, we see strong potential in deepening the integration of LLM-based reasoning with algorithmic search. By embedding structured computation within a language-guided planning process, agents can explore solution spaces more effectively. Extending this to coordinated multi-agent fine-tuning — with shared representations and task-aware interaction — could further strengthen system-level coherence and adaptability.

Finally, systems like KompeteAI may prove valuable beyond competitive AutoML settings. While platforms like Kaggle offer practical, open-ended challenges, they only partially reflect the demands of real-world scientific discovery. As autonomous agents begin contributing to hypothesis generation, experimental design, and even paper drafting, refining these tools for real research workflows becomes an exciting and urgent frontier.

## B  LIMITATIONS

Despite significant improvements over existing multi-agent AutoML solutions, our system still inherits several common limitations:

1. **Metric limitation:** Our approach is restricted to tasks with a single, well-defined numerical metric for evaluating solution quality. This limits applicability to domains where such a metric exists or can be reliably constructed.

2. **Data dependency:** Performance depends on how well the training data represents the test distribution. If the sample does not reflect the test set accurately, improvement over time is hampered. This is especially pronounced with distribution shifts or limited data.

3. **Benchmark scope:** Evaluation was conducted only on the Lite subset of MLE-Bench, the only comprehensive benchmark currently available. While Lite is limited, it correlates strongly with the full dataset (Pearson $r = 0.9670$, $p = 3.0 \times 10^{-10}$), providing a reliable proxy for overall performance.

Addressing these limitations is an important direction for future work, including extending the system to handle qualitative objectives and incorporating robust learning under distributional uncertainty.

## C  RELATED WORK DETAILS

### C.1  RAG

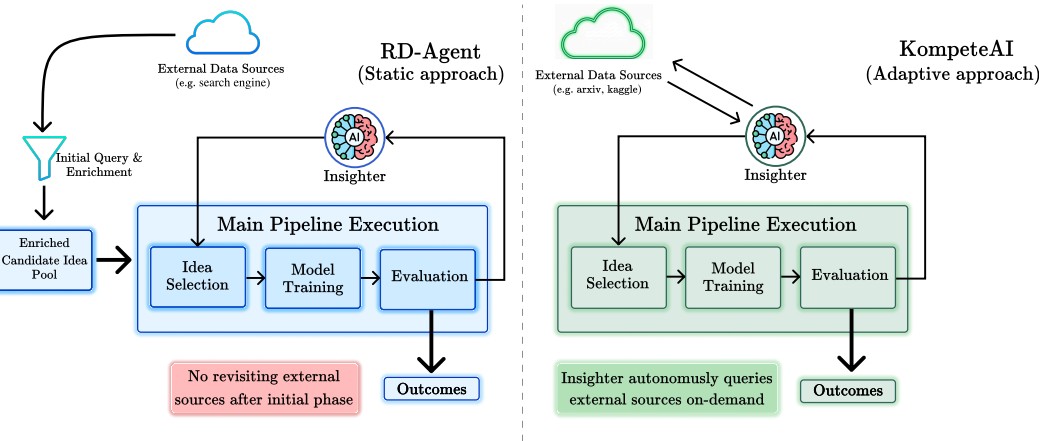

Figure A4: Comparison of external data integration in automl systems (RD-Agent vs. KompeteAI).

The utilization of external data sources within AutoML systems has been explored in a limited fashion in prior work, most notably in RD-Agent and AutoML-agent. In both of these systems, the integration of external knowledge follows a similar conceptual approach: prior to the execution of the main pipeline, the system queries a search engine and enriches its pool of candidate ideas with information extracted from the retrieved sources (see Figure A4).

However, this approach has several limitations. First, it is inherently static. The external knowledge is incorporated only at the outset, and the system does not revisit external sources during subsequent stages of the pipeline. As a result, if the system exhausts its initial set of promising ideas, it lacks the flexibility to dynamically seek out additional knowledge. This can lead to suboptimal performance, particularly in complex or evolving problem domains where continuous access to up-to-date information is crucial.

In contrast, our proposed system introduces a more adaptive mechanism for leveraging external data. Specifically, the idea-generating agent (Insighter) is endowed with the autonomy to decide

when to query external sources for supplementary information. Rather than relying solely on a one-time enrichment phase, Insighter continuously monitors its own progress and, upon encountering a creative impasse or a lack of effective solutions, proactively seeks out new ideas from the arXiv and Kaggle. This dynamic, on-demand integration of external knowledge enables the system to overcome stagnation and adapt to emerging challenges.

## C.2 MERGING

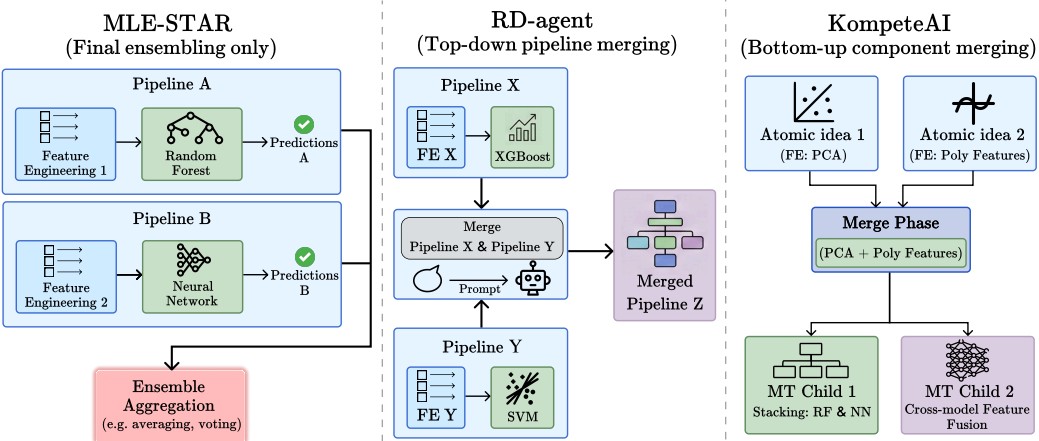

Figure A5: Comparison of merging strategies in LLM-based AutoML. Prior methods combine pipelines only through coarse or late-stage ensembling, whereas our approach performs fine-grained, multi-level merging of FE and MT components throughout the search, enabling systematic discovery of stronger hybrid configurations.

A key challenge in LLM-based AutoML is not only generating promising candidates, but effectively combining them into stronger configurations during the search. Prior systems such as RD-AGENT and MLE-STAR support only limited recombination, typically applied either at the end of the process or at a granularity too coarse to guide meaningful structural exploration. As illustrated in Figure A5, our approach replaces this with a principled, multi-level merging operator used throughout the search, allowing the agent to systematically combine complementary ideas at both the feature-engineering (FE) and model-training (MT) levels.

MLE-STAR performs merging only as a final ensembling step. This output-level aggregation is computationally simple but misses cases where individually weak components become effective when merged at the representational or architectural level. By deferring merging to the end, the system cannot explore intermediate combinations or guide the search trajectory, so synergistic FE or model structures never emerge, and suboptimal candidates may enter the final ensemble due to noisy validation scores. Merging thus becomes a post-hoc smoothing step rather than an active mechanism for solution improvement.

RD-AGENT attempts earlier recombination, but only by selecting high-scoring complete pipelines and prompting the LLM to merge them directly. This top-down process misses beneficial interactions between components that individually score poorly but work well together. Moreover, merging entire pipelines at once is a coarse operation: the LLM must reconcile complex architectures in a single step, often producing brittle or incoherent results. Because RD-Agent encodes stochastic heuristics (such as $\varepsilon$-greedy mixing) purely through text instructions, the LLM must implement algorithmic logic through free-form generation, which leads to inconsistent merges and limited control over structural outcomes.

In contrast, our system constructs pipelines from simple atomic ideas and increases their complexity through a merging phase applied throughout the search. By merging FE and MT components directly, rather than full pipelines, the agent can detect synergistic interactions early and exploit them systematically. Merged FE nodes spawn new MT children, and top-performing subtrees are resampled from each parent, allowing the system to combine impactful components while still exploring

---

**Algorithm 1** Adding Stage at Iteration $t$

---

**Input**: Ideation tree $T_t = (V_t, E_t)$; context vector $\mathbf{c}_t$
**Parameters**: $N$ (number of FE nodes), $M$ (number of MT nodes per parent)
**Output**: Updated tree $T_{t+1}$
1: $\mathbf{c}_t \leftarrow \mathbf{c}_t + \phi_{\text{EDA}}(\texttt{QueryEDA}(T_t))$
2: $\mathbf{c}_t \leftarrow \mathbf{c}_t + \phi_{\text{ext}}(\texttt{QueryExternal}())$
3: $\{v_j^{\text{FE}}\}_{j=1}^N \sim q_{\text{FE}}(\cdot \mid \mathbf{c}_t)$
4: **for** each $v_j^{\text{FE}}$ **do**
5:    $\{u_{j,i}^{\text{MT}}\}_{i=1}^M \sim q_{\text{MT}}(\cdot \mid v_j^{\text{FE}})$
6:    Assign scores $\{a_{j,i}\}_{i=1}^M$
7: **end for**
8: $T_t \leftarrow \texttt{Backpropagation}(T_t)$
9: Transform scores:
$$s_{j,i} = \begin{cases} a_{j,i}, & \text{if higher is better} \\ -a_{j,i}, & \text{otherwise} \end{cases}$$
10: Compute probabilities:
$$p_{j,i} = Softmax(s_{j,i})$$
11: Sample subset: $\mathcal{S} \sim \texttt{Sample}(\{u_{j,i}^{\text{MT}}\}, \{p_{j,i}\})$
12: **for** each $u \in \mathcal{S}$ **do**
13:    $\{w_k^{\text{MT}}\}_{k=1}^M \sim q_{\text{MT}}(\cdot \mid u)$
14: **end for**
15: $T_{t+1} \leftarrow \texttt{Backpropagation}(T_t)$
16: **return** $T_{t+1}$

---

new structures. This bottom-up process enables sophisticated configurations such as stacking, cross-model feature fusion, and novel multimodal architectures — structural innovations unattainable in MLE-STAR or RD-AGENT.

# D ALGORITHMS

## D.1 ADDING

This subsection formalizes the *adding* stage, as defined in Algorithm 1. We define the ideation tree at iteration $t$ as $T_t = (V_t, E_t)$, and the global context vector as $\mathbf{c}_t \in \mathcal{C}$, where $\mathcal{C}$ denotes the space of structured semantic representations.

The agent samples new Feature Engineering (FE) and Model Training (MT) nodes from conditional distributions:
$$q_{\text{FE}} : \mathcal{C} \rightarrow \mathcal{P}(\mathcal{V}_{\text{FE}}), \quad q_{\text{MT}} : \mathcal{V}_{\text{FE}} \rightarrow \mathcal{P}(\mathcal{V}_{\text{MT}})$$
where $\mathcal{V}_{\text{FE}}$ and $\mathcal{V}_{\text{MT}}$ are the respective candidate spaces for FE and MT nodes, and $\mathcal{P}(\cdot)$ denotes the space of probability distributions over a discrete set.

We also define a scoring function $a : \mathcal{V}_{\text{MT}} \rightarrow \mathbb{R}$, and a softmax-based selection mechanism used to stochastically expand the MT subtrees. The resulting tree $T_{t+1}$ is obtained by applying the additions and structural updates defined by the algorithm.

## D.2 MERGING

This subsection formalizes the *merging* stage, as defined in Algorithm 2. Let
$$\mathcal{M}_{\text{short}}, \mathcal{M}_{\text{long}} \subseteq \mathcal{P}(V_t^{\text{FE}})$$
denote short- and long-term memory buffers tracking merge success.

---

**Algorithm 2** Merging Stage at Iteration $t$

---

**Input**: Ideation tree $T_t = (V_t, E_t)$; memories $\mathcal{M}_{\text{short}}, \mathcal{M}_{\text{long}}$
**Parameters**: $N$ (number of FE merge candidates), $M$ (MT children per merged FE), $\theta_{\text{fail}}$ (failure threshold)
**Output**: Updated tree $T_{t+1}$, updated memories

1: $\{(v_i^{\text{FE}}, v_j^{\text{FE}})\}_{i=1}^N \sim q_{\text{pair}}(\cdot \mid T_t, \mathcal{M}_{\text{long}})$

2: **for** each $(v_i^{\text{FE}}, v_j^{\text{FE}})$ **do**

3:    $v_{ij}^{\text{FE}} \leftarrow \texttt{MergeFE}(v_i^{\text{FE}}, v_j^{\text{FE}})$

4:    $\{u_{ij,k}^{\text{MT}}\}_{k=1}^M \sim q_{\text{MT}}(\cdot \mid v_{ij}^{\text{FE}})$

5:    $\mathcal{S}_i, \mathcal{S}_j \sim \texttt{SampleTop}(v_i^{\text{FE}}, v_j^{\text{FE}})$

6:    Attach $\{u_{ij,k}^{\text{MT}}\} \cup \mathcal{S}_i \cup \mathcal{S}_j$ to $v_{ij}^{\text{FE}}$

7:    $\Delta_{ij} \leftarrow \texttt{Evaluate}(v_{ij}^{\text{FE}})$

8:    **if** $\sum_{r=1}^R \mathbb{I}\big[\texttt{IsFailure}(\Delta_{ij}^{(r)})\big] \geqslant \theta_{\text{fail}}$ **then**

9:       $(v_i^{\text{FE}}, v_j^{\text{FE}}) \in \mathcal{M}_{\text{long}}$

10:    **else**

11:       **if** $\texttt{IsFailure}(\Delta_{ij})$ **then**

12:          $(v_i^{\text{FE}}, v_j^{\text{FE}}) \in \mathcal{M}_{\text{short}}$

13:       **else**

14:          $(v_i^{\text{FE}}, v_j^{\text{FE}}) \in \mathcal{M}_{\text{long}}$

15:       **end if**

16:    **end if**

17: **end for**

18: $\mathcal{V}_{\text{FE}}^{\text{merge}} \sim q_{\text{FE}}(\cdot \mid T_t)$

19: **for** each $v^{\text{FE}} \in \mathcal{V}_{\text{FE}}^{\text{merge}}$ **do**

20:    $(u_p^{\text{MT}}, u_q^{\text{MT}}) \sim q_{\text{pair}}(\cdot \mid \texttt{Children}(v^{\text{FE}}))$

21:    $u_{pq}^{\text{MT}} \leftarrow \texttt{MergeMT}(u_p^{\text{MT}}, u_q^{\text{MT}})$

22:    Attach $u_{pq}^{\text{MT}}$ to $v^{\text{FE}}$

23: **end for**

24: $T_{t+1} \leftarrow \texttt{Backpropagation}(T_t)$

25: **return** $T_{t+1}, \mathcal{M}_{\text{short}}, \mathcal{M}_{\text{long}}$

---

# E KOMPETE-BENCH

## E.1 BENCHMARK DESCRIPTION

| | 0-99 Teams | 100-249 Teams | 250-999 Teams | 1000+ Teams |
|---|---|---|---|---|
| **Bronze** | Top 40% | Top 40% | Top 100 | Top 10% |
| **Silver** | Top 20% | Top 20% | Top 50 | Top 5% |
| **Gold** | Top 10% | Top 10 | Top 10 + 0.2%* | Top 10 + 0.2%* |

Table 6: Thresholds for winning a medal in Kaggle competitions vary depending on the number of teams participating in each competition. We implement the same thresholds as in MLE-bench. *the threshold increases by 1 for every 500 additional teams.* Source: Kaggle (2024).

Full descriptions of the competitions in Kompete-bench, including the name, number of participants, metrics, and medal thresholds, are listed in Table 6. The benchmark comprises 26 Kaggle

competitions, totaling 10.2 GB in size, and is divided into two distinct parts. The first part includes competitions from MLE-Bench that remain open for submissions on Kaggle and are each under 1 GB. These span from 2014 to 2017 and primarily feature straightforward tasks, where strong leaderboard positions can be achieved without complex modeling or novel ideas — sometimes even leveraging tools and libraries unavailable at the time of the original competition.

Given the intense competition in all recent featured Kaggle competitions, securing a top position has become exceedingly challenging. Therefore, in selecting datasets, we prioritized competitions that were both as recent as possible — to minimize the risk of data leakage into LLM training sets — and had a small memory footprint (under 3 GB per dataset). We manually curated these competitions from Kaggle, ensuring that solution downloads were still available for each selected challenge.

The distribution of competition end dates is illustrated in Figure A6, clearly showing the temporal separation between the two parts of the benchmark.

For all competitions, we report the percentage of participants outperformed ("percent users beaten") as the primary metric. This choice is motivated by the observation that current AutoML systems are unable to achieve medal positions in the latest competitions, yet tracking progress on these challenging tasks remains crucial. The "percent users beaten" metric is computed using the actual private Kaggle leaderboard, averaged over three independent runs. If a system fails to generate a valid submission, a score of 0% is assigned for that run. For ongoing competitions, evaluation is performed simultaneously on the public leaderboard.

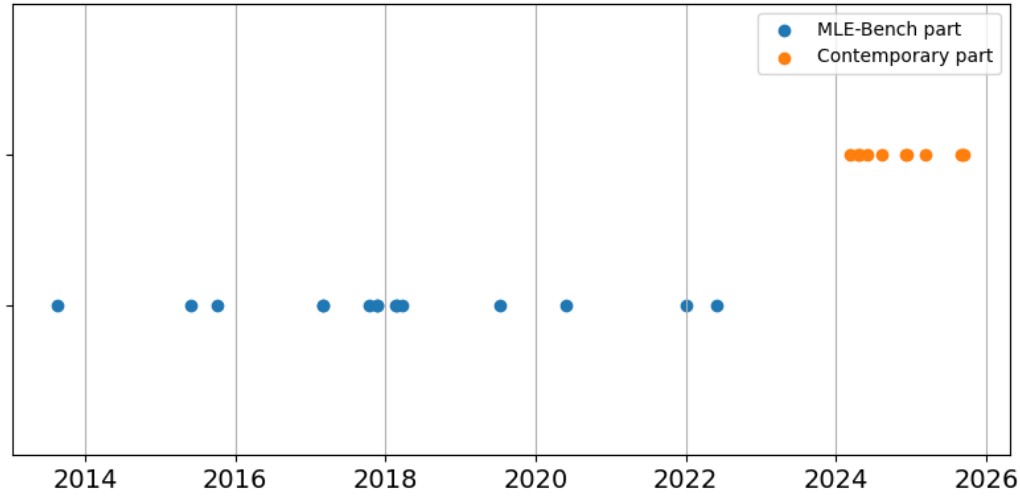

Figure A6: End dates for competitions presented in Kompete-Bench.

### E.2 CONFIDENCE INTERVALS

To ensure the robustness and reproducibility of our experimental results, we incorporated confidence intervals into our evaluation metrics. This approach addresses the inherent randomness.

Following the methodology established in MLE-Bench, we compute the confidence intervals using the standard formula: mean_score $\pm$ std.

where the mean and standard deviation are calculated over the aggregate metric scores obtained from three independent runs of each experiment.

Table 8 presents the evaluation metrics along with their corresponding confidence intervals.

| Name | Number of participants | Metric | Bronze | Silver | Gold | Part |
|---|---|---|---|---|---|---|
| aerial-cactus-identification | 1221 | ROC-AUC ↑ | 1 | 1 | 1 | MLE-bench (Lite) |
| denoising-dirty-documents | 162 | RMSE ↓ | 0.04517 | 0.02609 | 0.01794 | MLE-bench (Lite) |
| dog-breed-identification | 1281 | log loss ↓ | 0.04598 | 0.00539 | 0.0005 | MLE-bench (Lite) |
| dogs-vs-cats-redux-kernels-edition | 1315 | log loss ↓ | 0.06127 | 0.05038 | 0.03882 | MLE-bench (Lite) |
| jigsaw-toxic-comment-classification-challenge | 4539 | mean col-wise ROC AUC ↑ | 0.98639 | 0.98668 | 0.98740 | MLE-bench (Lite) |
| leaf-classification | 1596 | log loss ↓ | 0.01526 | 0.00791 | 0.00000 | MLE-bench (Lite) |
| mlsp-2013-birds | 81 | ROC-AUC ↑ | 0.87372 | 0.90038 | 0.93527 | MLE-bench (Lite) |
| nomad2018-predict-transparent-conductors | 879 | RMSLE ↓ | 0.06582 | 0.06229 | 0.05589 | MLE-bench (Lite) |
| plant-pathology-2020-fgvc7 | 1318 | ROC-AUC ↑ | 0.97361 | 0.97465 | 0.97836 | MLE-bench (Lite) |
| random-acts-of-pizza | 462 | ROC-AUC ↑ | 0.6921 | 0.76482 | 0.97908 | MLE-bench (Lite) |
| spooky-author-identification | 1242 | log loss ↓ | 0.29381 | 0.26996 | 0.16506 | MLE-bench (Lite) |
| tabular-playground-series-dec-2021 | 1189 | ROC-AUC ↑ | 0.95658 | 0.95658 | 0.9566 | MLE-bench (Lite) |
| tabular-playground-series-may-2022 | 1152 | ROC-AUC ↑ | 0.99818 | 0.99822 | 0.99823 | MLE-bench (Lite) |
| text-normalization-challenge-english-language | 261 | accuracy ↑ | 0.99038 | 0.99135 | 0.99724 | MLE-bench (Lite) |
| text-normalization-challenge-russian-language | 163 | accuracy ↑ | 0.97592 | 0.98232 | 0.99012 | MLE-bench (Lite) |
| eedi-mining-misconceptions-in-mathematics | 1449 | MAP@25 ↑ | 0.46090 | 0.49136 | 0.56429 | Contemporary |
| learning-agency-lab-automated-essay-scoring-2 | 2708 | quadratic weighted kappa ↑ | 0.83471 | 0.83518 | 0.83583 | Contemporary |
| lmsys-chatbot-arena | 1688 | log loss ↓ | 1.00472 | 0.99410 | 0.98392 | Contemporary |
| pii-detection-removal-from-educational-data | 2049 | efficiency score ↑ | 0.95714 | 0.95883 | 0.96615 | Contemporary |
| um-game-playing-strength-of-mcts-variants | 1610 | RMSE ↓ | 0.43050 | 0.42973 | 0.42591 | Contemporary |
| llm-prompt-recovery | 2176 | Sharpened Cosine Similarity ↑ | 0.6375 | 0.6513 | 0.6848 | Contemporary |
| equity-post-HCT-survival-predictions | 3327 | C-index ↑ | 0.69288 | 0.69320 | 0.69500 | Contemporary |
| cmi-detect-behavior-with-sensor-data | 2156 | F1 ↑ | 0.84 | 0.84 | 0.86 | Contemporary |
| make-data-count-finding-data-references | 833 | F1 ↑ | 0.548 | 0.564 | 0.620 | Contemporary |
| neurips-open-polymer-prediction-2025 | 1539 | wMAE ↓ | 0.057 | 0.041 | 0.032 | Contemporary |
| wsdm-cup-multilingual-chatbot-arena | 890 | categorization accuracy ↑ | 0.696381 | 0.702772 | 0.711412 | Contemporary |

Table 7: Summary of tasks and metrics included in Kompete-bench: the table lists competition names, number of participants, evaluation metrics, threshold values for medals and the part of the benchmark to which this competition belongs.

Table 8: Multi-agent AutoML systems comparison on KompeteAI-Bench with confedence intervals.

| | KompeteAI | w/o RAG | w/o merging | w/o scoring model | ML-Master | MLE-STAR | RD-Agent | AIDE |
|---|---|---|---|---|---|---|---|---|
| MLE-Bench part | $47.6 \pm 1.5$ | $45.0 \pm 1.1$ | $38.5 \pm 1.7$ | $41.6 \pm 1.4$ | $41.2 \pm 0.8$ | $37.3 \pm 1.1$ | $35.0 \pm 1.0$ | $32.5 \pm 1.4$ |
| Contemporary part | $14.0 \pm 1.1$ | $11.6 \pm 1.2$ | $9.5 \pm 1.0$ | $7.4 \pm 1.5$ | $10.4 \pm 0.8$ | $7.6 \pm 1.3$ | $6.9 \pm 1.5$ | $3.1 \pm 0.9$ |

# F    SETUP DETAILS

We used the hyperparameter settings summarized in Table 9, Table 10, Table 11. For AIDE and the RD-agent, we retained their default configurations as specified in the original implementations except for the time limit. For KompeteAI, we empirically selected a set of hyperparameters that strike a balance between the quality of component exploration and the computational time allocated to each. This tuning was guided by the need to ensure efficient coverage of critical components while maintaining tractable execution time. Below are the hyperparameters and their descriptions for each system.

## F.1    KOMPETEAI

**Hyperparameters tuning recommendation** Our system is characterized by a substantial number of hyperparameters-approximately 70 in total. From this extensive set, we identified 16 hyperparameters that are likely to have a significant impact on system performance, but were not explored in the ablation study of the main paper. These hyperparameters primarily influence the validation process, RAG, and adding and merging mechanisms.

While we did not conduct a comprehensive hyperparameter sensitivity analysis due to high complexity, the values reported in Table 9 were found to yield the most optimal results during the development phase. It is important to note that the optimal configuration of certain hyperparameters can be highly task-dependent. For instance, in complex research scenarios that require domain-specific knowledge, increasing the values of retrieve_n_papers and number_rag_ideas can be particularly beneficial.

In the following section, we provide detailed descriptions of each significant hyperparameter. This is intended to clarify their individual roles and to facilitate reproducibility and further research.

**Hyperparameters list** The list of hyperparameters used when running KompeteAI on benchmarks is given in Table 9.The time_run_minutes parameter sets the maximum runtime for the entire multi-agent system in minutes, after which the process will terminate. The runtime_error_time defines the time limit (in minutes) after which a generated code will be stopped. The subset_size_in_percent

specifies the percentage of the dataset to be used for quick validation. The validator_size_threshold sets a threshold for the dataset size; if the data exceeds this value, a subset is used for training and validation. The number_of_ideas_eda determines how many exploratory data analysis (EDA) ideas are generated per iteration. Similarly, number_of_ideas_data and number_of_ideas_modelling control the number of ideas related to feature engineering and model training. The max_add_idea parameter limits how many new ideas can be added to the idea pool in a single adding iteration. The number_of_selected_node specifies how many nodes are selected for adding expansion at each step. The number_of_iterations_parents sets how many iterations parent nodes participate in generating new ideas, while number_of_selected_node_merging determines how many nodes are chosen for merging at each iteration. The number_of_ideas_min and number_of_ideas_max define the minimum and maximum number of ideas, which are used as anchor ideas for the scoring model. The retrieve_n_papers and retrieve_n_competitions parameters control how many papers are retrieved from arXiv papers and Kaggle solutions. The number_rag_ideas sets how many ideas are generated using RAG. The memory_size parameter determines how many recent ideas, solutions, or states each agent remembers for learning and decision-making. Alternatively, if memory_size is set to nearest_nodes, the agent's memory consists of the most similar nodes or ideas, rather than a fixed number, allowing for more contextually relevant recall.

| Hyperparam name | Value |
|---|---|
| time_run_minutes | 360 |
| runtime_error_time | 30 |
| subset_size_in_percent | 10 |
| validator_size_threshold | $10^4$ |
| number_of_ideas_eda | 5 |
| number_of_ideas_data | 2 |
| number_of_ideas_modelling | 2 |
| max_add_idea | 2 |
| number_of_selected_node | 2 |
| number_of_iterations_parents | 2 |
| number_of_selected_node_merging | 2 |
| number_of_iterations_children | 3 |
| number_of_ideas_min | 2 |
| number_of_ideas_max | 5 |
| retrieve_n_papers | 3 |
| retrieve_n_competitions | 3 |
| number_rag_ideas | 5 |

Table 9: Hyperparameters used for running KompeteAI.

## F.2   RD-AGENT

The list of hyperparameters used when running RD-agent on benchmarks is given in Table 10. The debug_timeout parameter sets the maximum time, in seconds, that is allowed to debug one generated code. The full_timeout parameter defines the overall time limit for the system. The if_action_choosing_based_on_UCB flag determines whether the agents select their actions using the Upper Confidence Bound (UCB) strategy. The enable_knowledge_base flag indicates whether a shared knowledge base is enabled for the agents. The loop_n parameter specifies the number of main iterations the system will execute, set to 2000 to avoid early stopping not by the time limit. Finally, the max_trace_num parameter limits the number of traces that can be created during system execution.

| Hyperparam name | Value |
|---|---|
| debug_timeout | 3600 |
| full_timeout | 21600 |
| if_action_choosing_based_on_UCB | False |
| enable_knowledge_base | False |
| loop_n | 2000 |
| max_trace_num | 3 |

Table 10: Hyperparameters used for running RD-agent.

## F.3 AIDE

The list of hyperparameters used when running RD-agent on benchmarks is given in Table 11. The steps parameter defines the maximum number of steps the entire system can perform during a run. The max_debug_depth specifies the maximum depth for recursive code debugging. The debug_prob parameter suggests that debugging is enabled for every generated code, meaning that all relevant information will be recorded without any sampling. Finally, the time_limit parameter represents the maximum allowed wall-clock time (in seconds) for the experiment.

| Hyperparam name | Value |
|---|---|
| steps | 2000 |
| max_debug_depth | 20 |
| debug_prob | 1 |
| time_limit | 21600 |

Table 11: Hyperparameters used for running AIDE.

## G METRICS

| Competition | RD-agent | AIDE | ML-Master | MLE-STAR | KompeteAI |
|---|---|---|---|---|---|
| aerial-cactus-identification | 58 | 5 | 58 | 88 | 79 |
| denoising-dirty-documents | 3 | None | 15 | 0 | 12 |
| dog-breed-identification | 37 | 62 | 54 | 44 | 58 |
| dogs-vs-cats-redux-kernels-edition | 86 | 6 | 47 | 84 | 91 |
| jigsaw-toxic-comment-classification-challenge | 40 | 26 | 47 | 30 | 88 |
| leaf-classification | 33 | 24 | 35 | 54 | 46 |
| mlsp-2013-birds | None | 0 | 9 | 0 | None |
| nomad2018-predict-transparent-conductors | 78 | 33 | 76 | 30 | 22 |
| plant-pathology-2020-fgvc7 | 12 | 62 | 45 | 38 | 68 |
| random-acts-of-pizza | 58 | 47 | 56 | 58 | 70 |
| spooky-author-identification | 67 | 47 | 60 | 58 | 63 |
| tabular-playground-series-dec-2021 | 17 | 22 | 26 | 13 | 29 |
| tabular-playground-series-may-2022 | 16 | 32 | 29 | 23 | 43 |
| text-normalization-challenge-english-language | 3 | 9 | 25 | 7 | 27 |
| text-normalization-challenge-russian-language | 17 | 14 | 36 | 33 | 18 |

Table 12: Full table by "percent humans beaten" for AutoML systems given in the article at each competition in MLE-Bench part of Kompete-bench.

| Competition | RD-agent | AIDE | ML-Master | MLE-STAR | KompeteAI |
|---|---|---|---|---|---|
| cmi-detect-behavior-with-sensor-data | None | None | None | None | None |
| eedi-mining-misconceptions-in-mathematics | 19 | None | 18 | 9 | 30 |
| equity-post-HCT-survival-predictions | None | None | None | None | None |
| learning-agency-lab-automated-essay-scoring-2 | 9 | 23 | 15 | 17 | 13 |
| llm-prompt-recovery | 11 | 11 | 17 | None | None |
| lmsys-chatbot-arena | None | None | 14 | 17 | 15 |
| make-data-count-finding-data-references | 13 | None | 17 | 21 | 18 |
| neurips-open-polymer-prediction-2025 | 9 | None | 10 | 11 | 21 |
| pii-detection-removal-from-educational-data | None | None | 19 | 13 | 38 |
| um-game-playing-strength-of-mcts-variants | None | None | None | None | None |
| wsdm-cup-multilingual-chatbot-arena | 14 | None | 8 | 10 | 19 |

Table 13: Full table by "percent humans beaten" for AutoML systems given in the article at each competition in Contemporary part of Kompete-bench.

Results of the AutoML system on Kompete-bench are presented in Table 12 and Table 13. We report results using the "percent humans beaten" metric, which reflects the percentage of Kaggle leaderboard participants outperformed by each system. For every competition, each system was evaluated over three independent runs; the final score is the arithmetic mean of these runs, rounded to the nearest integer. If a system failed to generate any valid solution across all three attempts, its score is reported as None (equivalent to 0 when averaging across competitions). For individual runs where no valid submission was produced, a score of 0 was assigned. Notably, the primary factor influencing overall performance was the proportion of valid submissions: systems that consistently generated correct code achieved substantially higher scores.

# H ANALYSIS

## H.1 PERFORMANCE ASSESSMENT OF THE SCORING MODEL

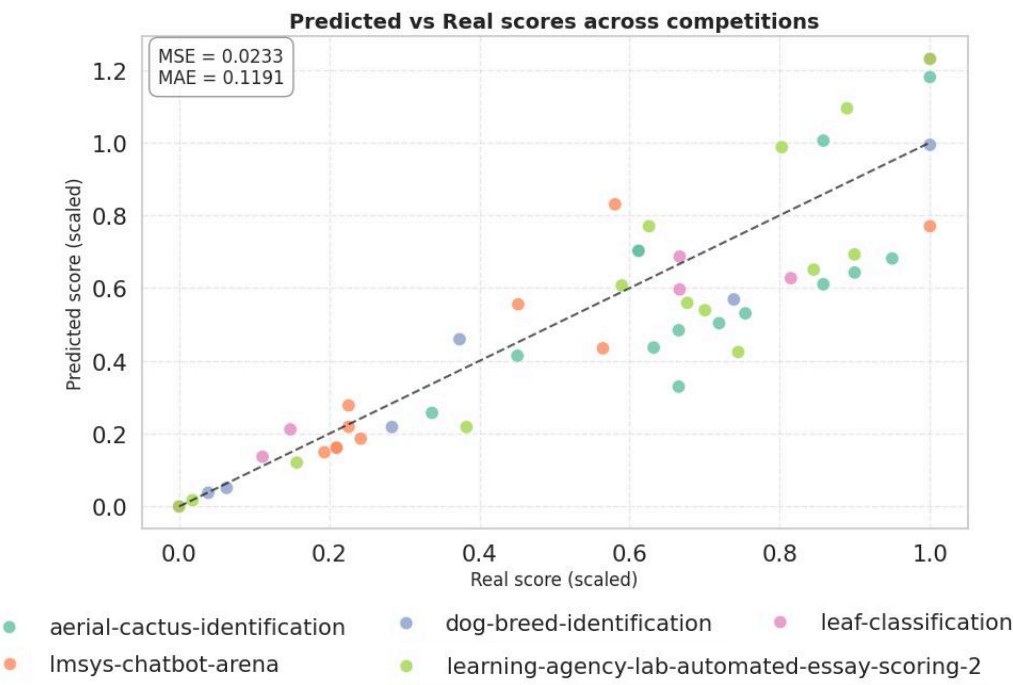

Figure A7: Comparison between predicted scores and actual validation scores after min–max normalization within each competition. Each point corresponds to a solution from a distinct competition. The solid black diagonal represents the ideal case where predicted and actual scores are perfectly aligned.

To evaluate the proposed scoring model, we conducted experiments across multiple benchmark competitions. Figure A7 compares predicted performance estimates with actual scaled scores obtained from validation splits of the data, covering tasks from image recognition to NLP and structured data. To account for the fact that competitions differ substantially in their metric ranges, we normalize scores within each competition using min–max scaling:

$$s_i^{\text{scaled}} = \frac{s_i^{\text{val}} - \min_j s_j^{\text{val}}}{\max_j s_j^{\text{val}} - \min_j s_j^{\text{val}}},$$

where $s_i^{\text{val}}$ denotes the validation score of solution $i$ within a competition. This adjustment ensures that relative differences are preserved while eliminating distortions caused by heterogeneous metric ranges, which would otherwise make cross-task comparisons less interpretable.

The results show a clear positive correlation between predicted and real scores. The model not only distinguishes strong solutions from weaker ones but also preserves relative ordering across a wide range of performance levels, which is critical for ranking-based exploration. At the same time, deviations from the diagonal indicate systematic tendencies: predictions for top-performing submissions are slightly conservative, while weaker candidates are occasionally overestimated. This asymmetry suggests that the model captures the general performance landscape but remains imperfect in calibrating extreme cases. Importantly, such biases are less problematic for the intended use case, where relative ranking — rather than precise absolute accuracy — guides the search process.

To further quantify ranking fidelity, Table 14 reports three standard order-preservation measures: NDCG, Spearman correlation, and Kendall's $\tau$, computed separately for each benchmark and aggregated across all tasks. These metrics evaluate the consistency between predicted and empirical rankings while being insensitive to absolute scale. Formally, NDCG is defined as

$$\text{NDCG} = \frac{1}{\text{IDCG}} \sum_{i=1}^{n} \frac{2^{y_i} - 1}{\log_2(i + 1)},$$

where $y_i$ denotes the ground-truth relevance at rank $i$ and IDCG is the value of the expression under the ideal ordering. Spearman correlation measures rank agreement via

$$\rho = 1 - \frac{6 \sum_{i=1}^{n} d_i^2}{n(n^2 - 1)},$$

where $d_i$ is the difference between predicted and true ranks of item $i$. Kendall's $\tau$ evaluates the fraction of correctly ordered pairs:

$$\tau = \frac{C - D}{C + D},$$

with $C$ and $D$ denoting the numbers of concordant and discordant pairs, respectively. High values across all three criteria confirm that the scoring model maintains strong relative ordering within and across competitions, even in cases where absolute predictions exhibit conservative bias.

Table 14: Ranking performance of the scoring model across benchmark competitions. NDCG evaluates the quality of the full predicted ranking with position-dependent weighting, while Spearman and Kendall $\tau$ assess rank-order consistency. Higher values indicate better agreement with ground-truth performance orderings.

| Competition | NDCG | Spearman | Kendall $\tau$ |
|---|---|---|---|
| aerial-cactus-identification | 0.977 | 0.758 | 0.690 |
| dog-breed-identification | 0.976 | 0.964 | 0.905 |
| leaf-classification | 0.961 | 0.900 | 0.800 |
| lmsys-chatbot-arena | 0.921 | 0.929 | 0.810 |
| learning-agency-lab-automated-essay-scoring-2 | 0.991 | 0.868 | 0.744 |
| **Overall** | 0.984 | 0.876 | 0.729 |

Two caveats remain: prediction errors may accumulate during long-running searches, and the evaluation is restricted to Kaggle-style benchmarks. Extending validation to real scientific discovery tasks represents an important future direction.

## H.2 IMPACT OF COMPETITION AND LITERATURE RETRIEVAL IN RAG

| Configuration Variant | Relative Performance (%) |
|---|---|
| Without competition knowledge | 86.7 |
| Without arXiv knowledge | 92.2 |
| Without any RAG input | 83.4 |

Table 15: Ablation of retrieval-augmented knowledge sources on a subset of benchmark competitions. Reported values indicate the relative performance by competition metric of each ablated configuration compared to the complete RAG-enabled system (treated as the 100% reference point).

The ablation analysis in Table 15 shows how different retrieval sources contribute to overall system performance. Removing competition data reduces relative accuracy to 86.7%, suggesting that competition write-ups often contain concrete heuristics and task-specific tricks that transfer effectively to our benchmark problems.

Excluding arXiv papers has a milder effect (92.2%). This is consistent with the idea that publications tend to capture broader methodological advances. While such knowledge is not always directly applicable to competition-style tasks, it still provides useful algorithmic patterns that enhance performance.

The largest drop is observed when retrieval augmentation is disabled entirely (83.4%). This indicates that external knowledge, regardless of source, is a key component of the approach. Together, competition reports and academic papers complement each other: the former supply practical, domain-oriented techniques, while the latter contribute generalizable methods and perspectives.

## H.3 DISTRIBUTION OF SUCCESSFULLY INTEGRATED RESEARCH IDEAS

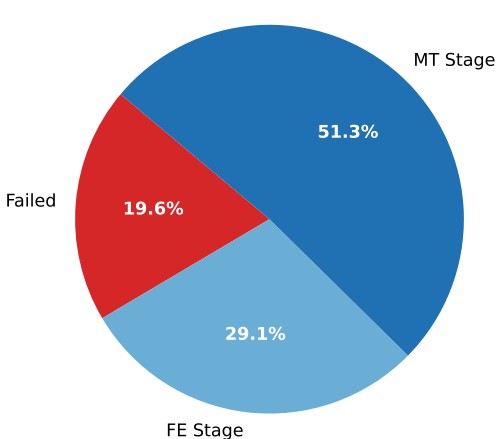

Figure A8: Average outcomes of the debugging process. The chart illustrates the proportion of successfully integrated ideas, categorized by their target AutoML stage — Model Training (MT) or Feature Engineering (FE) — versus the proportion of ideas that were ultimately discarded as "Failed".

Figure A8 summarizes the outcomes of the debugging and integration pipeline. Overall, 80.4% of proposed ideas were successfully incorporated into the research tree, confirming the reliability of debugging mechanisms. Only 19.6% of ideas failed to pass validation, that there are significantly fewer ideas that our system successfully debug.

## H.4 RELATIVE PERFORMANCE OVER TIME ANALYSIS

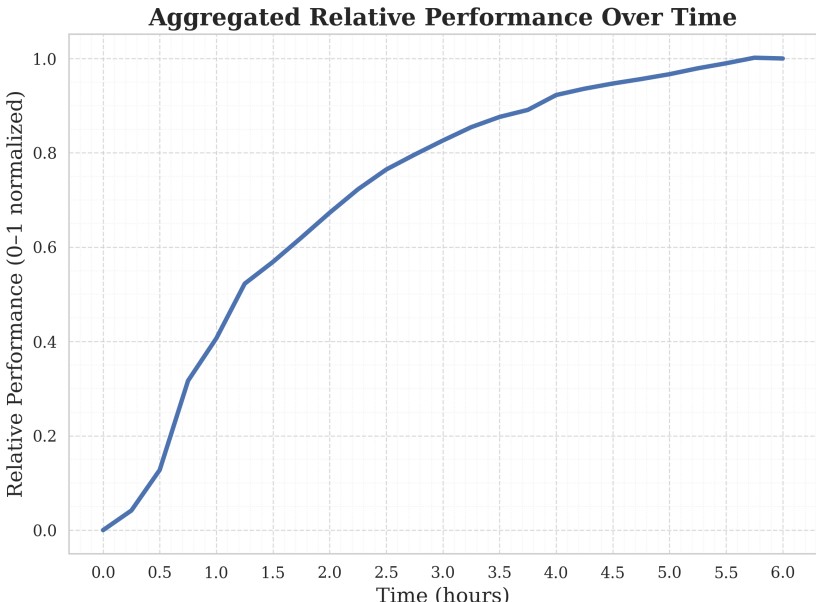

Figure A9: Aggregated Relative Performance Over Time. The curve represents the relative performance (normalized to the $0 - 1$ range, where $0$ is initial and $1$ is final performance) of KompeteAI's generated configurations over a six-hour competitive search window. The data is aggregated across multiple competitions/runs.

The temporal evolution of the aggregated relative performance is illustrated in Figure A9. In the initial phase, performance grows only modestly, reflecting the system's exploration of atomic ideas. This is followed by a pronounced surge as merging and adding operators are applied for the first time, enabling the generation of more complex solutions. Thereafter, performance increases more gradually, driven primarily by the merging of previously discovered ideas, which yields incremental improvements. Superimposed on this smooth trend, the curve exhibits a smaller but noticeable jump corresponding to the incorporation of additional knowledge through RAG during the adding phase. Over time, the score continues to rise steadily, approaching a near-asymptotic trend. It should be noted, however, that this does not imply the system has reached a true plateau: further gains are expected as merging and RAG operations continue to leverage newly generated ideas, particularly those emerging from subsequent research, though these improvements may require additional time.

## I LLM USAGE STATEMENT

We used Large Language Models exclusively as writing assistants to improve grammar, refine phrasing, and enhance the clarity and readability of the manuscript. LLMs were not involved in generating research ideas, designing the methodology, conducting experiments, analyzing results, or forming the scientific conclusions of this work. All conceptual and technical contributions were developed solely by the authors.

