# OpenReview forum: "KompeteAI: Accelerated Autonomous Multi-Agent System for End-to-End Pipeline Generation for Machine Learning Problems"
_ICLR.cc/2026/Conference — Submitted to ICLR 2026_

### Official Review · Reviewer_Ahze · 2025-10-26

**Soundness:** 3
**Presentation:** 3
**Contribution:** 3
**Rating:** 6
**Confidence:** 4

**Summary:**

The paper introduces KompeteAI, a multi-agent AutoML system for end-to-end ML pipeline generation. It tackles two major challenges faced by LLM-based AutoML frameworks—limited exploration and slow execution. KompeteAI employs two core operators, adding and merging, to dynamically expand and recombine strong pipeline components while leveraging Retrieval-Augmented Generation (RAG) to infuse real-world knowledge from Kaggle and arXiv. To mitigate execution bottlenecks, the system integrates a predictive scoring model for early performance estimation and an accelerated debugging paradigm using simplified code. Experiments on MLE-Bench and a newly proposed Kompete-bench show consistent state-of-the-art performance, with a 6.9× speedup and 3% higher accuracy over leading baselines like RD-Agent and ML-Master. The paper also highlights deficiencies in existing benchmarks and offers Kompete-bench as a fairer alternative reflecting real-world competition dynamics.

**Strengths:**

- The paper presents a well-structured and technically detailed framework that integrates algorithmic tree search with multi-agent coordination. The design of adding and merging operators is conceptually sound, addressing a real gap in exploration diversity and solution recombination that prior MCTS-based systems overlook.
- The introduction of a predictive scoring model and fast debugging loop represents a practical advance for efficiency; the ablation results clearly show how these components contribute to scaling search depth within a fixed runtime.
- The construction of Kompete-bench is another strong contribution. By identifying concrete flaws in MLE-Bench (data partition bias, leaderboard misalignment) and providing a curated, mixed-era benchmark with real Kaggle evaluation metrics, the authors make a valuable step toward more reproducible and realistic assessment of AutoML agents.

**Weaknesses:**

- Despite its solid engineering design, the overall conceptual novelty appears incremental. The core components—RAG integration, performance prediction, and tree-guided search—are thoughtful extensions of existing paradigms rather than fundamentally new algorithmic ideas. The contribution thus lies more in systematic synthesis and refinement than in conceptual breakthroughs.
- Although the experiments are extensive, they lack deeper behavioral analysis of the system. For instance, it remains unclear how merging strategies evolve during search, or under what conditions the predictive scoring model might introduce bias or mislead exploration. The evaluation primarily reports end metrics without qualitative insights into pipeline diversity, decision dynamics, or reasoning quality.
- The benchmark section, while valuable, does not clearly describe how the Contemporary subset was selected or how it concretely mitigates the evaluation bias inherent in MLE-Bench.
- The writing quality could be improved, as several sections contain minor typos and stylistic inconsistencies that slightly detract from readability.

**Questions:**

See #Weaknesses

---

> ### Author Response · Authors · 2025-11-24
>
> We thank you for your detailed and constructive feedback. Following your review, we have expanded several sections of the paper to better highlight the depth of our contributions and to address your concerns point by point.
>
> 1. You are correct that some components, such as RAG integration and tree-guided search, have been used in prior work. However, several aspects of our system introduce genuinely new paradigms to AutoML.
> First, regarding the predictive scoring model: while similar ideas exist in other domains, to the best of our knowledge we are the first to apply this paradigm within an AutoML framework. This component is essential for enabling deep search under strict time budgets. In the revised version, we provide a substantially more detailed technical explanation of the scoring model and include a new figure illustrating its mechanics (see line 296). This clarifies both how it works and what is novel about our design.
> Second, we introduce a new concept of memory-augmented merging, extending traditional recombination ideas by enabling the system to retain, reuse, and integrate strong pipeline fragments over time. To clarify the novelty relative to prior systems, we have added a detailed comparison and explanation in the Appendix (see line 708).
>
> 2. Thank you for encouraging a more thorough investigation of system behavior. We expanded the analysis in several directions:
> We added a dedicated Limitations section describing specific scenarios in which the system fails or behaves sub-optimally (see line 648). For each major component - RAG (see line 670), Merging (see line 708), and the Scoring Model (see line 1106), we now discuss failure modes, constraints, and potential sources of bias.
> In the Analysis section (see line 1104), we provide a deeper examination of how ideas evolve during search, how RAG data affects exploration, and how the scoring model influences decision trajectories.These additions give a more transparent and nuanced picture of system behavior beyond aggregate metrics.
>
> 3. We have substantially expanded the Kompete-Bench section to clearly explain how the Contemporary subset was constructed, why these tasks are representative of modern competition dynamics, and how the benchmark design explicitly mitigates known biases in MLE-Bench. The revised description (see line 850) now provides a unified and detailed justification of both the dataset selection process and the evaluation approach. We also provide a description of all competitions included in the Contemporary subset (see line 918)
>
> 4. Thank you for the comment. We have reviewed several aspects of our paper, including the Related Work, Scoring Model, and other sections. Additionally, we have thoroughly checked all chapters for grammatical, punctuation, and other types of errors, and have revised the text to improve its readability.
> Once again, thank you for the insightful review. Your feedback has helped us significantly improve the clarity, depth, and rigor of the paper.

---

### Official Review · Reviewer_i6sY · 2025-11-01

**Soundness:** 2
**Presentation:** 1
**Contribution:** 2
**Rating:** 2
**Confidence:** 4

**Summary:**

The paper proposes a system called KompeteAI, which is an autonomous multi-agent framework aimed at generating end-to-end machine-learning pipelines in a self-driving manner. The key features described include: A stage-decomposed multi-agent architecture, where agents handle tasks such as data ingestion, feature engineering, model training, hyperparameter tuning, and evaluation. A predictive scoring model that prunes weak solutions early (to reduce waste) and accelerates the pipeline generation process. An accelerated debugging paradigm, leveraging “simplified code and smaller data samples” to shorten the feedback loop during pipeline generation.
They also introduce a new benchmark called Kompete‑Bench to address limitations in MLE-Bench (e.g., reuse of training data for test sets).

**Strengths:**

1. Introduction of a benchmark (Kompete-Bench) is a positive step toward standardized evaluation in this space.
2. The multi-agent decomposition is intuitive, making the system design modular and potentially extensible.

**Weaknesses:**

1. Only a limited benchmark (MLE-Bench → Kompete-Bench) is used.
2. The empirical results do not demonstrate any statistically meaningful improvement. Across benchmarks, the reported mean scores overlap with baselines once standard deviations are considered. KompeteAI is 51.5 ± 1.5 while baseline is  48.2 ± 2.5.
3. Statistical rigor is weak. no mention of multiple random seeds, standard deviations, confidence intervals.
4. Presentation is not good. figure readability and caption clarity suffer; some key results are aggregated rather than broken down.

**Questions:**

1. What types of pipelines or datasets cause the system to fail or underperform? What are the resource costs (compute, memory) vs baseline? Without this, deployment risk is less clearly assessed.
2. Could the authors provide an ablation study isolating the contributions of (a) stage-decomposed architecture, (b) predictive scoring model, (c) accelerated debugging loop? Which component contributes most to speed or accuracy gains?
3. What are the resource costs (GPU hours, memory) compared to baseline methods? Is there a trade-off that practitioners should be aware of?

---

> ### Author Response · Authors · 2025-11-24
>
> We thank you for your thoughtful feedback. We appreciate your recognition of our contributions, particularly the introduction of Kompete-Bench and the modular multi-agent architecture. Below, we address your main concerns and questions.
>
> 1. You raised a valid concern regarding the limited benchmarking scope. Currently, MLE-Bench is the de facto standard for evaluating AutoML systems due to its broad coverage of diverse, curated competitions. The majority of baseline systems we compare against have only been evaluated on MLE-Bench, and re-running all baselines on alternative benchmarks is prohibitively resource-intensive.
> 2. Thank you for emphasizing the importance of rigorous statistical analysis. We would like to clarify that our evaluation has always included measures of variability and robust experimental protocols. In response to your comments, we have further strengthened the analysis:
>
> a) Updated Baselines: As of September 29, one of the baselines with previously overlapping confidence intervals has updated their results (see MLE-Bench commit: https://github.com/openai/mle-bench/commit/4b628fb52c777e23be89592e327c8df17911d6a5), and the overlap no longer exists.
>
>
> b) Statistical Significance: While our evaluation previously reported confidence intervals and averages over multiple runs, we have now conducted paired t-tests to explicitly assess the statistical significance of our improvements. These results are reported in Table 3 (line 432–442) and confirm that our system achieves statistically significant gains over the baselines.
>
>
> c) Consistency Across Benchmarks: Confidence intervals (mean ± std) for all results on Kompete-Bench continue to be reported in the same manner as for MLE-Bench, ensuring consistent statistical reporting.
>
>
> d) Experimental Protocol: As before, for MLE-Bench each system was executed three times per competition with different random seeds, and averages were reported; similarly, for Kompete-Bench, results are averaged over three runs per competition. Detailed information on the evaluation environment is provided in the Environment subsection (line 391) and in the captions of the corresponding tables and figures (lines 429 and 501).
>
>
> e) Extended Analysis: To provide a more complete picture of performance variability, additional statistical details are available in the Appendix (line 1065).
>
> These clarifications show that our empirical improvements are both statistically meaningful and reproducible, addressing concerns about overlapping scores and statistical rigor without changing the overall evaluation methodology.
>
> 3. We have revised several figures and captions to improve readability and clarity. If you could specify which diagrams remain problematic, we would be grateful and will address them in the next revision.
>
> 4. You asked about failure cases. We have added a dedicated Limitations section (Appendix, line 648) discussing scenarios where KompeteAI underperforms. In particular, the system struggles when it is not possible to construct a representative validation set or when a suitable numerical metric for the task cannot be defined. We are actively exploring ways to mitigate these limitations in future work.
>
> 5. You asked about resource usage. On MLE-Bench, KompeteAI was evaluated using a single A100 40GB GPU for 6 hours per task (MLE-Bench default: 1 A10 24GB, 24 hours per task; however, most baseline systems are actually run under less constrained conditions — for example, ML-Master uses an A100 40GB GPU and 12 hours, while RD-Agent is typically run on a V100 with 24 hours). On Kompete-Bench, all systems were run under identical conditions: 1 A100 40GB GPU and 6 hours. In practice, resource consumption is more dependent on the specific task than on the system itself. Our observations indicate that the GPU is rarely fully utilized by any solution generated by our system. See line 391 for details.
>
> 6. Thank you for suggesting a more detailed ablation study. However, due to the tightly coupled nature of our stage-decomposed architecture, it is not meaningful to remove certain components in isolation. For example, without the stage-decomposed approach, the pipeline generation process degenerates into a shallow tree that halts prematurely, failing to utilize the full time budget in most cases. Similarly, the merging stage is only effective when paired with the adding stage, as new ideas must be generated before they can be merged. The predictive scoring model also relies on the accelerated debugging loop to ensure code validity. Nevertheless, our ablation study (see line 486) demonstrates that the merging stage, in particular, significantly boosts performance, and we have clarified these dependencies in the text.
>
> Thank you again for your constructive feedback, which has helped us improve the clarity, rigor, and completeness of our work. We hope that these revisions and clarifications address your concerns, and we look forward to any further suggestions you may have

---

### Official Review · Reviewer_umEn · 2025-11-01

**Soundness:** 3
**Presentation:** 3
**Contribution:** 3
**Rating:** 6
**Confidence:** 4

**Summary:**

The paper introduces KompeteAI, an innovative multi-agent AutoML system designed to address the exploration and execution limitations of Large Language Model (LLM)-based AutoML frameworks. In the Introduction, the authors outline challenges with current methods, including poor iterative refinement, constrained solution recombination, and slow code validation. Related Work highlights prior AutoML architectures and the need for dynamic knowledge integration and efficient scoring. KompeteAI’s architecture, detailed in Section 3, partitions the pipeline into modular stages managed by specialized agents, leverages dynamic Retrieval-Augmented Generation (RAG) from external sources, and introduces adding and merging operators for compositional exploration. A novel scoring model and accelerated debugging paradigm significantly reduce execution time. Section 4 presents Kompete-bench, a new benchmark curated to fairly evaluate multi-agent AutoML systems using recent and diverse Kaggle competitions. Experiments in Section 5 demonstrate that KompeteAI exceeds state-of-the-art results on MLE-Bench and Kompete-bench, with ablation studies confirming the critical impact of RAG, merging, and fast scoring. The Conclusion affirms KompeteAI’s advancements in automated pipeline generation, flexible knowledge integration, and robustness in real-world ML challenges.

**Strengths:**

- Multi-agent, tree-guided exploration with explicit mechanisms for “adding” and “merging” ideas/pipeline components is novel, directly addressing existing limitations in MCTS and one-shot LLM-based AutoML.
- The use of dynamic, contextually adaptive RAG—retrieving and integrating external SOTA solutions in pipeline generation—extends current practices and enhances autonomy.
- Extensive, comparative experiments against SOTA baselines under fixed resource constraints; ablations elucidate the contribution of each major component (RAG, merging, scoring model).
- The Kompete-bench benchmark, curated with care to avoid data leakage and to align more honestly with actual human-level performance, deepens the impact of the empirical study.
- Results are reported with mean ± SEM and badge/percentile-based metrics, reflecting both robustness and practical relevance.
- Demonstrated acceleration (6.9× iteration speedup) with little to no decrease in quality.
- The overall motivation, system overview, and empirical design are clearly articulated.
- Improved performance (e.g., +3% average on MLE-Bench vs. SOTA) and clearer benchmarking protocols directly benefit both academic research and practical AutoML deployment.

**Weaknesses:**

- More information is needed on how merging ensures improvements, how “structural and statistical traits” are reconciled, and how the merging operator avoids destructive or trivial recombinations.
- The architecture, few-shot setup, and validation protocol of the scoring model are only briefly described. Metrics around its recall/precision in “early stopping” and error rates—critical for safe pruning—are not fully reported.
- Sensitivity analysis of system hyperparameters (memory buffer sizes, thresholds for merging failures, RAG sample numbers) is lacking, so the method’s stability and robustness are unclear.
- Guidance for tuning or default values for these hyperparameters is not given.
- While mean ± SEM is reported, there is no explicit mention of statistical significance testing (e.g., paired t-test), even though some reported differences fall within the error range.
- The number of seeds/runs and randomization details for ablation studies are unclear in the main text, making it hard to judge reliability.
- Known failure cases (e.g., where KompeteAI fails on “Contemporary” competitions) are only implied (“far behind top leaderboard teams”) rather than explicitly analyzed or illustrated with examples.
- Resource usage breakdown (latency, memory) and limitations (e.g., when dynamic RAG or merging could backfire, or nonscalability) are not thoroughly discussed.

**Questions:**

- Can the authors clarify the architecture of the predictive scoring model and how it is trained, particularly regarding few-shot anchor selection and prevention of overfitting to anchor tasks The scoring model is pivotal in accelerating evaluation, but operational specifics—especially training data selection and validation—are deferred to brief descriptions and may affect both bias and validity.

- How does the adaptive RAG component interact with the rest of the system at inference time, particularly in settings where access to external sources (like Kaggle notebooks) is unavailable or potentially restricted? The generalizability and fairness of KompeteAI rest on the ability to source knowledge consistently; if RAG behavior degrades in restricted environments, it could affect both the utility and reproducibility of the framework.

- Can you provide, in the main text, pseudocode or a concise step-by-step algorithm for both the “adding” and “merging” operations, not just refer to the appendix? The current high-level description lacks concrete, reproducible detail; providing explicit algorithms would clarify the method's implementation and assist reproducibility.*

- How is the scoring model validated for reliability in early-stopping poor candidates versus missing late-emerging good solutions, and what metrics define its predictive quality? The effectiveness (and risk of bias) of the scoring model could fundamentally affect search coverage; reporting quantitative metrics (e.g., error or recall rates for early pruning) is critical for evaluating practical safety/performance trade-offs.*

- What hyperparameter sensitivity analyses (e.g., memory buffer sizes, thresholds for merging failures, number of RAG samples) were performed, and how stable is system performance to these changes? Stability with respect to these design choices affects both reliability and adoption; results in the main text or at least brief summary (even if details are in appendix) are necessary.

- Can you clarify what statistical tests (if any) were conducted to affirm the significance of observed performance improvements (e.g., KompeteAI’s +3% over ML-Master on MLE-Bench)? The tables report mean ± SEM, but no explicit significance testing is mentioned; clarification is needed to interpret whether the reported gains are robustly significant.

- Could you provide more details on the ablation study design (e.g., number of seeds/runs per system per condition, randomization protocol)? While ablation results are presented, the experimental protocol (number of repeats, control of random factors) is not fully clear, which is crucial to evaluating variance and the reliability of these component-wise claims.

---

> ### Author Response · Authors · 2025-11-24
>
> We thank you for the thorough and constructive feedback on our work. We appreciate the recognition of KompeteAI’s novel multi-agent design, dynamic RAG integration, modular pipeline architecture, and the extensive empirical evaluation on both MLE-Bench and Kompete-Bench. Below, we address each of your comments in detail, providing clarifications, additional experimental details, and further explanations regarding the scoring model, merging and adding operations, hyperparameter sensitivity, and statistical analyses.
>
> 1. You raised an important point regarding the mechanics of the merging operator and how it ensures genuine improvements rather than trivial recombinations. In the revised version of the paper, we have expanded the description of the merging approach and explicitly compared it with similar concepts found in other systems like RD-Agent and MLE-STAR (see line 708). Regarding the utility of this operator: while it is true that some recombinations can be trivial, the primary strength of merging lies in its ability to evolve more complex pipelines. By combining multiple distinct models or feature engineering ideas into a single workflow, the system can achieve significantly higher metric stability and performance than individual agents acting in isolation. At the same time, it is important to emphasize that the merging operator is not meant to introduce fundamentally new conceptual ideas. We expect that genuinely novel or nontrivial solution components - including those generated through RAG - are added through the adding operator. In contrast, the purpose of merging is to consolidate the strongest parts of existing solutions, combining complementary models, transformations, or partial pipelines into more robust composite architectures. Although merging can yield new architectural structures through recombination, these structures are not “new” in the conceptual sense; rather, they represent refined and synergistic integrations of previously discovered ideas.
>
> 2. We agree that the scoring model is a pivotal component that requires a more detailed explanation. We have updated the methodology section to provide a comprehensive description of its internal workings, including a new schematic diagram to improve readability and understanding of the flow (see line 295). To address your concerns about validation and early stopping, we have added a numerical analysis that explicitly reports the metrics used to evaluate the model’s predictive quality. In this analysis, we assess both the deviation from real performance values (using MAE) and the ranking effectiveness of predicted solutions (using NDCG), thereby ensuring that the pruning process preserves high-potential candidates while saving computational resources (see line 1106).
>
>
> 3. Regarding the sensitivity analysis for hyperparameters (such as memory buffer sizes), we acknowledge that this is a valuable aspect of system evaluation. However, performing a full, exhaustive sweep of these parameters is computationally prohibitive given the scale of the experiments. Furthermore, our development process revealed that the optimal values for many of these hyperparameters are highly task-dependent rather than universal. Nevertheless, to aid reproducibility and practical adoption, we have added a section in the updated paper that lists the standard default values we used, along with a guide offering tips on how to tune them for different scenarios (see line 955).
>
> 4. You correctly noted the need for explicit statistical testing to validate our performance gains. In response, we added paired t-tests to validate the results on MLE-Bench and introduced confidence intervals for Kompete-bench to ensure consistent statistical reporting across benchmarks. The corresponding t-test results are presented in Table 3 (line 432). Regarding the experimental setup, we have clarified our randomization protocol: for MLE-Bench, we executed our system three times per competition using different random seeds and reported the average number of medals; similarly, for Kompete-bench, each system was run three times, with results averaged across competitions. All details of the evaluation environment are provided in the Environment subsection (line 391), as well as in the captions of the corresponding tables and figures (lines 429 and 501). To give a more complete view of performance variability, we have also included an extended statistical analysis in the Appendix (line 1065).

---

> ### Author Response · Authors · 2025-11-24
>
> 5. To address your questions regarding resource consumption and system limitations, we have provided a clearer breakdown of our computational constraints. On MLE-Bench, KompeteAI utilizes a single A100 40GB GPU with a strict 6-hour time limit, which is significantly more constrained than the standard setting of A10 and 24 hours. On Kompete-Bench, all compared systems were given identical resources (1 A100, 6 hours) to ensure fairness. Our experiments suggest that memory and compute usage are often more dependent on the specific dataset and task complexity than on the AutoML framework itself. Finally, we have added a dedicated "Limitations" chapter in the Appendix (see line 648), where we discuss constraints—including the behavior of the system in restricted environments where dynamic RAG might be limited.
>
> 6. Thank you for this helpful suggestion. In the revised version, we provide concise step-by-step descriptions of both the “adding” and “merging” operations directly in the main text (see lines 251 and 274). These outlines enumerate all essential phases and actions needed to reproduce the method at the conceptual and procedural levels. We believe this level of detail strikes an appropriate balance between clarity and readability in the main body. The full pseudocode, which includes additional technical steps and implementation-level nuances, remains in the Appendix to avoid overloading the main text with low-level operational details. This separation allows readers to quickly understand the workflow while still giving practitioners access to the complete algorithmic specification when needed.
>
> 7. Thank you for raising this important point regarding the behavior of the adaptive RAG component in restricted environments. Due to the specific implementation of our RAG system, all knowledge is automatically collected by a dedicated script prior to competition — this is a general collection of ideas and resources, not specific to a given competition. During inference, the agent itself selects and forms the subset of ideas it will actually use, ensuring that only relevant information is applied to the current task.
> Potential limitations may arise when accessing sources such as arXiv, where real-time retrieval could be restricted. To address this, we conducted an additional analysis of agent performance under conditions where certain knowledge sources are unavailable (see line 1188). This ablation study shows that while the absence of competition knowledge or arXiv papers slightly reduces performance, the system remains largely robust, and the overall methodology is still reproducible. These results indicate that the adaptive RAG component contributes meaningfully to performance, but the framework’s design ensures resilience in scenarios with partial source availability.

---

### Official Review · Reviewer_753t · 2025-11-01

**Soundness:** 2
**Presentation:** 2
**Contribution:** 2
**Rating:** 4
**Confidence:** 2

**Summary:**

In this work, the authors introduce a new AutoML framework called KompeteAI. That is, a framework for automating machine learning tasks. A key idea of the approach is to decompose the machine learning workflow into separate stages, to be able to focus on specific tasks. Then, at a later stage, partial solutions are combined using an approach that involves adding and merging operations. The paper also introduces approaches to handle execution performance, as well as introduces a new benchmark called Kompete-bench.

**Strengths:**

- The general problem of improving AutoML

- The benchmarking shows promising results compared to recent LLM-based AutoML frameworks, both when comparing with different (leading board) LLMs and with the same underlyingLLM (Figure 2)

**Weaknesses:**

- In the related work section, the competing frameworks, such as MLE-STAR and RD-AGENT, are strongly criticized, e.g., "producing incoherent or suboptimal integrations" and "simply consolidates ideas without meaningfully improving solution exploration.", but the justification for such strong statements is not that clear.

- Certain parts of the paper are clearer and more precisely described than others. E.g., the section about the scoring model is fairly easy to get a high-level idea, but lacks technical depth.

- The paper becomes a bit unfocused, trying to convey both a new framework for AutoML, and to provide, motivate, and describe a new benchmark. I would recommend focusing on only one of these directions in one paper.

**Questions:**

- Please describe in more detail how your RAG approach (stated as dynamic in Table 1) compares to AutoML and  and RD Agent, that are stated to be in "R&D-phase". What is the justifications for this conclusion?

- Please describe more in depth how the controlled merger of this paper differs from the recombination of the RD approach.

---

> ### Author Response · Authors · 2025-11-24
>
> We thank you for your thoughtful and constructive feedback. We appreciate your recognition of the strengths of our work, particularly the benchmarking results and the relevance of the AutoML problem. Below, we address your comments and questions in detail.
>
> 1. We acknowledge your concern regarding the strong statements about competing frameworks such as MLE-STAR and RD-Agent in the related work section. In the revised version, we have carefully revisited this subsection (lines 96–104), updating the textual discussion as well as the description of the comparative table to ensure a more balanced and precise presentation. We have also clarified and substantiated our points by providing more concrete examples and additional diagrams in the Appendix. Specifically, we now detail the limitations observed in the integration and exploration strategies of these frameworks, supported by empirical evidence (see lines 670 and 708). Our intention is not to diminish prior work, but to clearly position our contributions and highlight the specific technical gaps that KompeteAI addresses.
>
>
>
> 2. Thank you for pointing out the need for a more in-depth description of the scoring model. In the revised paper, we have expanded this section to include a detailed explanation of the model’s architecture and evaluation metrics. We have also added a diagram to visually illustrate the scoring process (see line 295). If there are additional technical aspects that remain unclear, we would greatly appreciate further guidance on which sections could benefit from more detail.
>
>
> 3. We appreciate your suggestion regarding the focus of the paper. Our motivation for presenting both the KompeteAI framework and the Kompete-bench benchmark stems from the lack of lightweight, accessible benchmarks at the time of our development — MLE-Bench, for example, is resource-intensive and not suitable for rapid iteration. Moreover, in our analysis we show that the only existing benchmark exhibits a noticeable bias when evaluating systems relative to the real leaderboard, which further motivated the need to design a new, more representative benchmark. We agree that this dual focus may appear somewhat diffuse, and in the revision we have clarified the relationship between the framework and the benchmark, emphasizing how Kompete-bench was instrumental in the development and evaluation of KompeteAI. We will consider splitting these contributions in future work, but believe that, for this submission, their joint presentation is justified by their interdependence.
>
>
> We have made several revisions throughout the paper to improve clarity and technical depth, particularly in sections that were previously less precise. If there are specific areas that remain unclear, we would be grateful for further feedback.

---

### Meta-Review · Area_Chair_83z7 · 2026-01-06

**Summary:**

This work proposes KompeteAI, a multi-agent LLM-based AutoML framework that addresses limited exploration and execution bottlenecks. It introduces dynamic tree-based exploration with solution merging, integrates retrieval-augmented generation from real-world sources, and employs accelerated evaluation and debugging to reduce costly executions, achieving state-of-the-art performance on AutoML benchmarks.

**Reviewer Concerns:**

Key issues identified by reviewers: (1) Justification & Clarity: Strong criticisms of prior work (RD-Agent, MLE-STAR) lack evidence; merging operator and scoring model are insufficiently detailed. (2) Methodological Transparency: Architecture, training, few-shot anchor selection, RAG integration, and pseudocode for adding/merging operations are unclear; safeguards against destructive recombination are not described. (3) Evaluation Rigor: Number of seeds/runs, statistical significance testing, and early-stopping reliability metrics are missing; reported improvements overlap with baselines. (4) Robustness & Sensitivity: Hyperparameter analysis, system stability, failure cases, and resource usage are underexplored. (5) Scope & Focus: The Paper conflates the new AutoML framework with a new benchmark; benchmark design and bias mitigation are insufficiently explained.

After the rebuttal, I think several concerns have not been addressed such as novelty and evaluation.

**Reviewer Scores:**

All reviewers would keep their score unchaged.

---

### Decision · Program_Chairs · 2026-01-26

Reject